# Fair Transit Stop Placement: A Clustering Perspective and Beyond

**Haris Aziz**[1]   **Ling Gai**[2]   **Yuhang Guo**[1]   **Jeremy Vollen**[3]

## Abstract

We study the transit stop placement (TrSP) problem in general metric spaces, where agents travel between source–destination pairs and may either walk directly or utilize a shuttle service via selected transit stops. We investigate fairness in TrSP through the lens of justified representation (JR) and the core, and uncover a structural correspondence with fair clustering. Specifically, we show that a constant-factor approximation to proportional fairness in clustering can be used to guarantee a constant-factor bi-parameterized approximation to core. We establish a lower bound of $1.366$ on the approximability of JR, and moreover show that no clustering algorithm can approximate JR within a factor better than $3$. Going beyond clustering, we propose the Expanding Cost Algorithm, which achieves a tight $2.414$-approximation for JR, but does not give any bounded core guarantee. In light of this, we introduce a parameterized algorithm that interpolates between these approaches, and enables a tunable trade-off between JR and core. Finally, we complement our results with an experimental analysis using small-market public carpooling data.

## 1. Introduction

A municipality has decided to offer a publicly-operated shuttle to offer its residents a safe, convenient, and accessible alternative to private vehicles. Towards this end, the public infrastructure planner has been allocated a budget to construct a desired number of shuttle stops. Given data describing the common trips made by each resident that would use the shuttle service, how can the planner decide where to place the shuttle stops? This problem, which we call the *Transit Stop Placement Problem* (TrSP), is the focus of this work.

While the focus of prior public transportation research in civil engineering and operations research is typically on efficiency and cost minimization (Desaulniers & Hickman, 2007; Miller et al., 2016; Teodorovic & Janić, 2016; Ceder, 2016), there has been a growing appreciation of equity concerns when designing and planning infrastructure in the last decade (Martens, 2016; Martens & Lucas, 2018; Bullinger et al., 2025). In terms of our problem, while each resident (referred to as agents, henceforth) would prefer to have a shuttle stop at the exact addresses of their starting point (e.g., home) and destination (e.g., work), the aim of the planner is to select a set of stops that is as fair as possible to the agents, subject to scarce resources. Bullinger et al. (2025) recently studied this question in the case in which all agents and potential stops are located on a line. Inspired by the literature on committee voting (see, e.g., Aziz et al., 2017; Lackner & Skowron, 2023), they defined fairness properties known as *core* and *justified representation* (a relaxation of core). Rather than take an egalitarian formulation of fairness, which may overcorrect for an isolated resident and hence result in stop placements which are not convenient for *any* agent, core and justified representation are based upon the ideal of proportional representation, i.e., one in which groups of agents with similar preferences are entitled to resources in proportion to their size.

The primary limitation of the work by Bullinger et al. (2025) lies in its focus on the line, which while a natural introductory setting for investigating fairness in transit stop placement, fails to adequately capture the complexity of many real-world transportation networks. In this work, we address this limitation by studying transit stop placement in general metric spaces.

To make things more concrete, we offer an example. Suppose there are six agents, and each agent $i$ follows a route from $a_i$ to $b_i$. The planner has a budget to construct three shuttle stops. Each agent derives disutility from a transit stop placement equal to the sum of the agent's walking distance to the stops they use and their transit cost between stops. There are four candidate stop locations, $c_1$ through $c_4$. Consider the stop placements depicted in Figure 1a and Figure 1b, where the selected stops are indicated by stars. We

[1]Department of Computer Science, University of New South Wales, Sydney, Australia [2]School of Business, University of Shanghai for Science and Technology, Shanghai, China [3]Department of Computer Science, Northwestern University, Chicago, USA. Correspondence to: Yuhang Guo <yuhang.guo2@unsw.edu.au>.

*Proceedings of the 43rd International Conference on Machine Learning*, Seoul, South Korea. PMLR 306, 2026. Copyright 2026 by the author(s).

observe that the placement in Figure 1a fails to adequately represent the agent group $\{1, 2, 3, 4\}$. Every agent in that group derives greater disutility from the transit stop placement in Figure 1a than from the placement in Figure 1b. This group constitutes two thirds of the population, and thus intuitively should have two thirds of the decision power, i.e. be able to effectively decide two of the three stops.

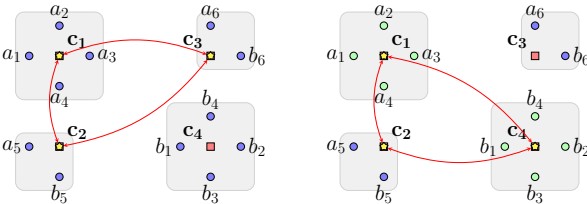

*(a)* Stop placement at $c_1, c_2, c_3$    *(b)* Stop placement at $c_2, c_3, c_4$

*Figure 1.* Transit stop placement example. Each travel route connects an agent pair $(a_i, b_i)$, with six agents in total (blue circles). There are four candidate stop locations, $c_1$ to $c_4$ (red squares). Panels (a) and (b) illustrate two different placement choices, with the selected stops marked by yellow stars. Red arrows indicate the shuttle transit routes.

In this work, we aim to identify algorithms which can provably guarantee fair transit stop placements in general metric spaces. Since the provably fair algorithms introduced by Bullinger et al. (2025) exploit the particular structure of the line, we must take a different approach. We note that our problem shares substantial overlap with fair centroid clustering (Chen et al., 2019; Micha & Shah, 2020; Caragiannis et al., 2024; Aziz et al., 2024; Kellerhals & Peters, 2024; Cookson et al., 2025), which also involves selecting a prescribed number of centers (or stops) for a given set of points while pursuing fairness guarantees. Indeed, the application of a fair clustering algorithm to our previous example would certainly require the selection of both $c_1$ and $c_4$. This observation naturally raises the question of whether existing fair clustering algorithms, backed by a rich body of research on fairness, can be adapted to achieve our fairness objectives in the context of transit stop placement.

> *To what extent can fair clustering algorithms guarantee proportionally representative and fair transit stop placements in general metric spaces?*

The two settings differ, however, in that agents are associated with a pair of points in our model whereas they are modeled as a single point in clustering, and thus clustering algorithms necessarily ignore potentially useful information on agents' preferences. Furthermore, clustering algorithms do not capture that agents' preferences may also depend on their transit cost between stops. We would like to define transit cost flexibly on instances defined on general metric spaces since walking and other transport infrastructure may

not necessarily align.[1]

> *Is it possible to devise algorithms which outperform clustering algorithms if we take a more fine-grained view of agents' preferences and flexibly account for agents' transit times?*

### 1.1. Our Contributions

Our first contribution is a flexible model of transit stop placement in general metric spaces. In the model of Bullinger et al. (2025), transit times between stops are assumed to be proportional to the walking distance between the two stops. Our model, at its most general, allows for significantly more general transit times, requiring only that they abide by a distance function which satisfies the triangle inequality. As was done in Bullinger et al. (2025), we define an approximate version of justified representation, $\beta$-JR, which requires that all agents in a deviating group prefer the deviation by a factor $\beta \geq 1$. We also define a bi-criteria approximation of core, $(\alpha, \beta)$-core, in which the $\alpha$ factor strengthens the deviating group size requirement.

We first establish formal connections between centroid clustering and transit stop placement (TrSP) under the assumption of negligible transit times. By reducing TrSP to clustering, we show that any clustering algorithm satisfying $\rho$-Proportional Fairness (PF) can be used to guarantee a $(2, \rho)$-core outcome for TrSP. Since the Greedy Capture algorithm gives a $(1 + \sqrt{2})$-PF solution in clustering, a direct corollary of our theorem is that this algorithm in the context of TrSP, which we refer to as GC-TrSP, guarantees $(2, 1 + \sqrt{2})$-core. We also prove that GC-TrSP satisfies $(2 + \sqrt{5}) \approx 4.236$-JR, and that both of these bounds are tight. In the reverse direction, we give a mapping from clustering instances to TrSP instances which demonstrates that any $\beta$-Justified Representation (JR) algorithm for TrSP can be used to obtain a $2\beta$-PF solution for the original clustering instance. This insight leads to an impossibility result: in general metric spaces, a JR (and therefore, core) outcome is not guaranteed to exist. We improve on this impossibility by showing that no TrSP algorithm can guarantee better than $\frac{\sqrt{3}+1}{2} \approx 1.366$-JR in general. To understand the limitations of our reduction to clustering with respect to JR, we construct a TrSP instance for which no clustering algorithm can guarantee better than 3-JR.

To surpass this barrier, we introduce a novel algorithm, the ***Expanding Cost Algorithm (ECA)***, which evaluates and selects stop *pairs* rather than singletons, and directly considers agents' costs rather than mere distances. We prove that ECA satisfies $(1 + \sqrt{2}) \approx 2.414$-JR, again with a tight bound, thereby improving on the approximation guarantee of every

---

[1]For example, a footbridge may allow an agent to move between two points on foot faster than other infrastructure allows.

clustering-based algorithm. Strikingly, this approximation factor holds for any travel times between stops that satisfy the triangle inequality. In contrast, no clustering algorithm guarantees a constant factor JR approximation under such generality. Although ECA substantially outperforms clustering algorithms in terms of JR approximation, it fails to satisfy $(\alpha, \beta)$-core for any $\alpha, \beta \geq 1$. A comprehensive picture of JR approximations is depicted in Figure 2 while the comparison of GC-TrSP, ECA, and $\lambda$-Hybrid with respect to core approximations is summarized in Table 1.

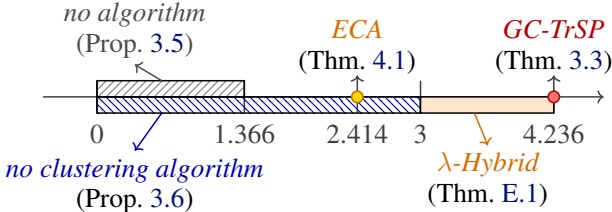

*Figure 2.* Overview of JR approximation ratios. The two shaded regions with diagonal lines indicate the lower bounds for general algorithms and clustering algorithms, respectively. The points at 2.414 and 4.236 correspond to the ECA and GC-TrSP algorithms. The performance of the $\lambda$-Hybrid algorithm ranges between 3 and 4.236, depending on the choice of the parameter $\lambda$.

| | **Core** | |
|---|---|---|
| | Lower Bound | Upper Bound |
| **GC-TrSP** | $(2, 1 + \sqrt{2})$ (Prop. 3.2) | $(2, 1 + \sqrt{2})$ (Prop. 3.2) |
| **ECA** | $(\gamma, \rho)$-core for any $\gamma, \rho \geq 1$ (Prop. 4.2) | – |
| $\lambda$-**Hybrid** | $\left(2, \frac{\sqrt{4\lambda^2 + 12\lambda + 1} + 2\lambda + 1}{4\lambda}\right)$ (Prop. E.4) | $\left(2, \frac{\sqrt{\lambda^2 + 6\lambda + 1} + \lambda + 1}{2\lambda}\right)$ (Cor. E.3) |

*Table 1.* Comparison of Greedy Capture for TrSP (GC-TrSP), Expanding Cost Algorithm (ECA), and $\lambda$-Hybrid in terms of approximation ratios for core in general metric space.

We defer some of our results to appendices. Given that GC-TrSP and ECA exhibit complementary strengths in approximating JR and the core, we propose a new algorithm, $\lambda$-Hybrid, which interpolates between GC-TrSP and ECA via a tunable parameter $\lambda \geq 0$ (Section E.2). We note that the algorithm by Bullinger et al. (2025), which provides JR in the line instance, is in fact also a clustering algorithm on the line, in which setting it guarantees a PF solution. However, it can perform arbitrarily poorly when the candidate centers do not align with data points. To address this, we propose a new algorithm called $\ell$-dictator partition algorithm, which generalizes the algorithm by Bullinger et al. (2025) and satisfies PF in clustering on the line, and thus matches their $(1, 2)$-core bound for TrSP on the line. Lastly, we prove that a solution minimizing the total cost for TrSP in general metric space is NP-hard to compute.

## 2. Preliminaries

### 2.1. Transit Stop Placement Model

For any $t \in \mathbb{N}$, let $[t] := \{1, 2, \ldots, t\}$. Let $\mathcal{X}$ be a set and $d$ and $d'$ be two distinct distance functions satisfying the triangle inequality. The metric space $(\mathcal{X}, d)$ represents *walking cost*[2] while the metric space $(\mathcal{X}, d')$ represents *transit cost*. An instance of the *Transit Stop Placement* (TrSP) model is defined by the tuple $\mathcal{I} = \langle N, \mathcal{C}, \{\theta_i\}_{i \in N}, k\rangle$, where $N = [n]$ is a finite set of $n$ agents, $\mathcal{C}$ is a set of $m$ candidate transit stops, $k$ is a positive integer, and $\theta_i = (a_i, b_i)$ denotes the *endpoints* between which agent $i$ travels, where $a_i, b_i \in \mathcal{X}$. We denote by $\Theta(S)$ the multiset of all endpoints associated with agents in a subset $S \subseteq N$. For simplicity, let $\Theta := \Theta(N)$. A solution to a TrSP instance is a subset of candidate transit stops $Y \subseteq \mathcal{C}$. The solution $Y$ is said to be *feasible* if $|Y| \leq k$. For ease of exposition, we define $d(i, X) = \min_{j \in X} d(i, j)$. Given any feasible solution $Y$, the cost of agent $i$ with type $(a_i, b_i)$ is given by $c_i(Y) =$

$$\min \left\{ d(a_i, b_i), \min_{y_1, y_2 \in Y} \left[ d(a_i, y_1) + d'(y_1, y_2) + d(y_2, b_i) \right] \right\}.$$

Intuitively, each agent $i$ minimizes her travel cost by either walking directly from $a_i$ to $b_i$, or by walking from $a_i$ to a transit stop $y_1 \in Y$, taking the transit system from $y_1$ to another stop $y_2 \in Y$, and then walking from $y_2$ to her destination $b_i$. Unless otherwise specified, we adopt the assumption that $d'(y_1, y_2) = 0$ for all $y_1, y_2 \in Y$. This assumption, which we refer to as *null transit cost*, captures scenarios where the cost of using the transit system is negligible compared to walking. Critically, our central algorithmic result will hold for arbitrary transit cost.

The total travel cost of a solution $Y \subseteq \mathcal{C}$ is defined as $c(Y) = \sum_{i \in N} c_i(Y)$. We first remark that a solution which minimizes total travel cost among all feasible solutions is NP-hard to compute, even in the special case of null transit times. Since we are mainly focused on fairness in this work, we defer the proof to Appendix B. This result marks a contrast between general metric spaces and the line, since the latter setting admits a polynomial time algorithm for any transit cost which are directly proportional to the walking cost (Bullinger et al., 2025).

**Proposition 2.1.** *Unless $P = NP$, there is no polynomial time algorithm which computes a minimum cost solution to the TrSP problem, even under null transit cost, i.e., even when $d'(i, j) = 0$, for all $i, j \in \mathcal{X}$.*

We focus on the fairness axioms introduced by Bullinger et al. (2025), both of which draw inspiration from the study of *core stability* in various domains including multi-winner voting, participatory budgeting, and fair clustering (see, e.g., (Fain et al., 2016; Pierczyński & Skowron, 2022; Chaudhury et al., 2022; Aziz et al., 2023; Chaudhury et al., 2024;

---

[2] A standard interpretation of the cost is travel time.

Peters, 2025)). In the context of transit stop placement, core is grounded in the principle that, given an instance with $n$ agents and a budget of $k$, each agent is entitled to a $\frac{k}{n}$-proportion of the budget. Toward this end, the core comprises the set of feasible imputations in which no coalition of agents can all strictly improve their cost by using their budget on an alternative set of transit stops. That is, a solution $Y$ is in the core if for any subset of agents $S \subseteq N$ and any transit stop set $T \subseteq \mathcal{C}$ with $|S| \geq |T| \cdot \frac{n}{k}$, there exists at least one agent $i \in S$ such that $c_i(T) \geq c_i(Y)$.

Our other fairness axiom of focus, *justified representation* (JR), weakens core by restricting its attention to deviations consisting of pairs of stops. That is, JR requires that any group of agents with size at least $\lceil \frac{2n}{k} \rceil$ should not be able to reduce their travel costs by deviating from solution $Y$ to an alternative pair of stops. A pair of stops is considered the minimal meaningful unit since no agent can derive benefit from a single stop. Besides being significantly easier to compute, JR has another major advantage over core. Whereas the complexity of checking core in our setting is unsettled, JR can be verified in polynomial time.[3]

As we will see, JR (and thus core) is not guaranteed to exist in our setting. As a result, we will study multiplicative relaxations of both properties, which we will now introduce. For approximate JR, we require that at least one agent in the group improve by no more than a factor $\beta$.

**Definition 2.2** ($\beta$-Justified Representation ($\beta$-JR)). A solution $Y \subseteq \mathcal{C}$ is said to provide $\beta$-JR if for every subset of agents $S \subseteq N$ with $|S| \geq \frac{2n}{k}$ and every pair of transit stops $T \subseteq \mathcal{C}, |T| = 2$, there exists an agent $i \in S$ such that $\beta \cdot c_i(T) \geq c_i(Y)$.

For core, we additionally parameterize the size of a deviating group of agents using multiplicative factor $\alpha$.

**Definition 2.3** (($\alpha, \beta$)-core). Let $\alpha, \beta \geq 1$. A solution $Y \subseteq \mathcal{C}$ is in the ($\alpha, \beta$)-*core* if for every subset of agents $S \subseteq N$ and every transit stop set $T \subseteq \mathcal{C}$ with $|S| \geq \alpha \cdot |T| \cdot \frac{n}{k}$, there exists an agent $i \in S$ such that $\beta \cdot c_i(T) \geq c_i(Y)$.

When $\beta = 1$, the ($\alpha, 1$)-core aligns with the core approximation defined by Bullinger et al. (2025), which represents a multiplicative *size approximation*, relaxing the requirement such that a group of agents can deviate and establish $|T|$ transit stops only if its size is at least $\alpha$ times the number of agents who "deserve" $|T|$ stops. Conversely, when $\alpha = 1$, the ($1, \beta$)-core introduces a relaxation on individual cost, aligning with the notion of approximate Proportional Fairness (PF) in fair clustering (Chen et al., 2019).

---

[3]For further details, see Appendix F or Bullinger et al. (2025, Appendix C.1).

## 2.2. Fair Clustering Model

We will now review the problem description and relevant definitions of fair centroid clustering. To avoid notation confusion with the TrSP instance, we use a slightly modified notation. Similar to TrSP, clustering instances consist of points in a metric space $(\mathcal{X}, d)$. A fair clustering instance is given by a tuple $\mathcal{I}' = \langle N', \mathcal{C}', k' \rangle$ where $N'$ is a finite set of $n'$ datapoints, $\mathcal{C}'$ is a set of $m'$ *centers*, and $k'$ is a positive integer. A clustering solution is a subset $P \subseteq \mathcal{C}'$ of at most $k'$ centers. We will draw and exploit connections between our fairness concepts and the clustering fairness concept known as *proportional fairness*.

**Definition 2.4** ($\rho$-Proportional Fairness ($\rho$-PF)). A clustering solution $P \subseteq \mathcal{C}'$ with $|P| \leq k'$ satisfies $\rho$-Proportional Fairness if, for all $S' \subseteq N'$ with $|S'| \geq \frac{n'}{k'}$ and for all $c \in \mathcal{C}'$, there exists a datapoint $i \in S'$ with $\rho \cdot d(i, c) \geq d(i, P)$.

Put differently, there should be no group of agents large enough to deserve one center that would all prefer that center to their closest center under $P$, even when scaling their alternative distance by $\rho$.

Lastly, we introduce a mapping from instances of TrSP to instances of fair clustering. Given any TrSP instance $\mathcal{I} = \langle N, \mathcal{C}, k, \{\theta_i\}_{i \in N} \rangle$, we define a clustering instance $\mathcal{I}^C = \langle \Theta, \mathcal{C}, k \rangle$ within the same metric space $(\mathcal{X}, d)$. We call this the *clustering instance induced by $\mathcal{I}$*, or simply induced clustering instance, when context is clear. In words, the clustering instance induced by $\mathcal{I}$ reinterprets the endpoints of agents in the TrSP model as datapoints in a clustering instance and maintains the same candidate set and target selection number. Notably, the set of feasible outcomes in the TrSP instance $\mathcal{I}$ is identical to that in its induced clustering instance $\mathcal{I}^C$. As a result, every clustering algorithm immediately yields an algorithm for TrSP instances by the following simple procedure: given a TrSP instance $\mathcal{I}$, run the clustering algorithm on $\mathcal{I}^C$ and return the output. In the next section, we will reason about the application of clustering algorithms to transit stop placement.

## 3. Transit Stop Placement Meets Clustering

In this section, we establish connections between the TrSP problem and fair clustering. We first show that clustering algorithms can be used to approximate fairness in our setting.

### 3.1. Approximate Fairness by Reduction to Clustering

To establish that fair clustering algorithms can indeed be used to guarantee fairness in TrSP, we prove a metatheorem which uses our reduction from TrSP to clustering under null transit cost assumption.

**Theorem 3.1.** *Given a TrSP instance $\mathcal{I}$, if $Y$ is a feasible solution satisfying $\rho$-PF in the induced clustering instance*

$\mathcal{I}^C$ for some $\rho \geq 1$, then $Y$ is $(2, \rho)$-core solution for $\mathcal{I}$.

*Proof.* Given any TrSP instance $\mathcal{I} = \langle N, \mathcal{C}, \{\theta_i\}_{i \in N}, k \rangle$, consider the induced clustering instance $\mathcal{I}^C = \langle \Theta, \mathcal{C}, k \rangle$ and let $Y$ be a feasible centroid selection satisfying $\rho$-PF. For any subset of agents $S \subseteq N$ and any subset of transit stops $T \subseteq \mathcal{C}$ with $|S| \geq |T| \cdot \frac{2n}{k}$ in $\mathcal{I}$, let $\Theta(S)$ denote the multiset of datapoints corresponding to agents in $S$. Then $|\Theta(S)| = 2 \cdot |S|$, and hence we have $\frac{1}{2} \cdot |\Theta(S)| \geq |T| \cdot \frac{2n}{k}$. Define the set $Q := \{j \in \Theta(S) : \rho \cdot d(j, T) \geq d(j, Y)\}$. Since the solution $Y$ satisfies $\rho$-PF in $\mathcal{I}^C$ and $|\Theta| = 2n$, it follows that for any candidate center $c \in T$, the number of datapoints $j \in \Theta(S)$ such that $\rho \cdot d(j, c) < d(j, Y)$ is strictly less than $\frac{2n}{k}$. Therefore, $|Q| > |\Theta(S)| - |T| \cdot \frac{2n}{k} \geq |\Theta(S)| - \frac{1}{2} \cdot |\Theta(S)| = \frac{1}{2} \cdot |\Theta(S)|$. Note that each datapoint in $Q$ is an agent's endpoint in $\mathcal{I}$ and $|Q| > \frac{1}{2} \cdot |\Theta(S)| = |S|$. By the pigeonhole principle, there exists an agent $i^* \in S$ such that both $a_{i^*}$ and $b_{i^*}$ belong to $Q$. Hence, it holds that $\rho \cdot d(a_{i^*}, T) \geq d(a_{i^*}, Y)$ and $\rho \cdot d(b_{i^*}, T) \geq d(b_{i^*}, Y)$. This tells us the following about the cost of agent $i^*$:

$$\rho \cdot c_{i^*}(T)$$
$$= \rho \cdot \min \left\{ d(a_{i^*}, b_{i^*}), \min_{\tau_1, \tau_2 \in T} [d(a_{i^*}, \tau_1) + d(b_{i^*}, \tau_2)] \right\}$$
$$= \min \{ \rho \cdot d(a_{i^*}, b_{i^*}), \rho \cdot d(a_{i^*}, T) + \rho \cdot d(b_{i^*}, T) \}$$
$$\geq \min \{ \rho \cdot d(a_{i^*}, b_{i^*}), d(a_{i^*}, Y) + d(b_{i^*}, Y) \}$$
$$\quad (\rho \cdot d(a_{i^*}, T) \geq d(a_{i^*}, Y); \rho \cdot d(b_{i^*}, T) \geq d(b_{i^*}, Y))$$
$$\geq \min \{ d(a_{i^*}, b_{i^*}), d(a_{i^*}, Y) + d(b_{i^*}, Y) \} \quad (\rho \geq 1)$$
$$= c_{i^*}(Y).$$

Therefore, for any subset of agents $S \subseteq N$ and any subset of transit stops $T \subseteq \mathcal{C}$ with $|S| \geq |T| \cdot \frac{2n}{k}$ in $\mathcal{I}$, there always exists an agent $i^*$ such that $\rho \cdot c_{i^*}(T) \geq c_{i^*}(Y)$. This implies that the solution $Y$ satisfies $(2, \rho)$-core in $\mathcal{I}$. $\square$

From Theorem 3.1, it immediately follows that $(2, 1 + \sqrt{2})$-core solutions for the TrSP problem can be computed using the clustering algorithm known as Greedy Capture (Chen et al., 2019). When applied to TrSP instances (through the reduction described in Section 2.2), we refer to this algorithm as Greedy Capture for TrSP (GC-TrSP).

The algorithm works by uniformly growing balls around candidate transit stops and iteratively adding stops whose balls capture a sufficient number of uncaptured endpoints. In more detail, each endpoint is first marked "active", and GC-TrSP smoothly increases radius $r$ and iteratively "opens" stops which are at distance at most $r$ from at least $\lceil \frac{2n}{k} \rceil$ active endpoints. Opened stops are added to the solution and endpoints are deactivated as soon as they are contained in an opened ball. The algorithm terminates when all agents are deactivated.[4] See Section C for formal pseudocode.

In fact, as we will now show, $(2, 1 + \sqrt{2})$-core is the best achievable approximation factor for GC-TrSP, and thus the analysis provided by Theorem 3.1 is tight. Instances proving tightness of approximation and several other proofs are deferred to Appendix D.

**Proposition 3.2.** *GC-TrSP satisfies $(2, 1 + \sqrt{2})$-core. However, for any $\delta, \varepsilon > 0$, there exists an instance for which GC-TrSP violates $(2 - \delta, 1 + \sqrt{2} - \varepsilon)$-core.*

One of the apparent drawbacks of the core approximation obtained by Greedy Capture is that it strengthens the coalition size requirement by a factor 2. This effectively halves each group's representative decision power when considering deviations. It turns out that the coalition size requirement *must* be strengthened to some extent in order for GC-TrSP to give any bounded guarantee with respect to core. In particular, it holds that GC-TrSP does not satisfy $(1, \rho)$-core for any $\rho \geq 1$.[5] As a natural next step, we investigate GC-TrSP with respect to JR, a property which maintains the proportional coalition size requirement but restricts considered deviations to those consisting of pairs of stops. The following result shows that GC-TrSP achieves a $(2 + \sqrt{5})$-approximation to JR, and this bound is tight.

**Theorem 3.3.** *GC-TrSP satisfies $(2 + \sqrt{5})$-JR. However, for any $\varepsilon > 0$, there exists an instance for which GC-TrSP violates $(2 + \sqrt{5} - \varepsilon)$-JR.*

*Proof Sketch.* Given any TrSP instance $\mathcal{I}$, let $Y \subseteq \mathcal{C}$ be the set of stops returned by GC-TrSP. Suppose that solution $Y$ violates $(2 + \sqrt{5})$-JR. Then there exists a group of agents $S \subseteq N$, a pair of stops $T = \{\tau_1, \tau_2\}$ such that $(2 + \sqrt{5}) \cdot c_i(T) < c_i(Y)$ for every $i \in S$. We will show that for the solution returned by GC-TrSP, there always exists an agent $j \in S$ such that $(2 + \sqrt{5}) \cdot c_j(T) \geq c_j(Y)$, contradicting to our assumption. We first define $r_T$ as the maximum distance between any endpoint of an agent in $S$ and two stops $\tau_1$ and $\tau_2$, i.e., $r_T = \max_{j \in S} \{\max\{d(a_j, \tau_1), d(b_j, \tau_2)\}\}$,[6] and denote by $i^*$ the agent who attains the maximum $r_T$.

We first claim that during the execution of GC-TrSP, the growing radius must reach $r_T$. Suppose GC-TrSP terminates at some radius strictly smaller than $r_T$, then agent $i^*$

---

[4] We note that Greedy Capture may terminate before selecting

$k$ centers, an artifact that appears in some of our lower bound arguments. This behavior can be avoided by deactivating exactly $2n/k$ (fractional) endpoints for each selected center. It is not clear whether the choice of which endpoints to deactivate can be used to improve the bounds. Nonetheless, we also give lower bound results like Proposition 3.6 which apply to all clustering algorithms.

[5] Consider an example with two agents and $k = 3$ on the unit interval. Suppose $\mathcal{C} = \{0, \frac{1}{4}, \frac{1}{2}, \frac{3}{4}, 1\}$ and the voters have endpoints $(0, \frac{1}{2})$ and $(0, 1)$. GC-TrSP selects $\{0, \frac{3}{4}\}$, causing each agent to incur a cost of $\frac{1}{4}$. Note that the solution $\{0, \frac{1}{2}, 1\}$ is feasible and gives each agent a cost of 0.

[6] Without loss of generality, we assume that every agent in $S$ travels from transit stop $\tau_1$ to $\tau_2$ under stop pair $T$.

would have $c_{i^*}(Y) \leq 2 \cdot r_T \leq 2 \cdot c_{i^*}(T)$, which contradicts the assumption. Henceforth, we focus on the case that GC-TrSP reaches radius $r_T$. We next distinguish cases by how many stops from $T$ belong to $Y$. (1). Assume $\tau_1 \in Y$ and $\tau_2 \notin Y$. By definition of $r_T$, $\tau_2$ can cover $\lceil \frac{2n}{k} \rceil$ endpoints with radius $r_T$. However, GC-TrSP does not select $\tau_2$, implying there is one endpoint of some agent $i' \in S$ already deactivated before reaching $r_T$ by some other stop $y \in Y$, which gives us $c_{i'}(Y) \leq d(a_{i'}, \tau_1) + r_T$. By considering the minimum multiplicative cost improvement of $i^*$ and $i'$, we bound the cost of $i^*$ by $\frac{3+\sqrt{13}}{2} \cdot c_{i^*}(T) \geq c_{i^*}(Y)$, contradicting that $(2 + \sqrt{5}) \cdot c_i(T) < c_i(Y)$ for each $i \in S$. (2). Both $\tau_1$ and $\tau_2$ are excluded from $Y$. Using the similar argument as above, we upper-bound the cost of three agents in $S$, including $i^*$, under solution $Y$ and upper-bound the multiplicative improvement by a ratio of $(2 + \sqrt{5})$, which finally completes the proof. $\square$

## 3.2. Fairness Lower Bounds

In this section, we contextualize the JR approximation obtained by GC-TrSP (Theorem 3.3) by establishing lower bounds on JR, both in terms of clustering algorithms and general existence. In contrast with the line metric, where JR is guaranteed to exist (Bullinger et al., 2025), we show that a solution satisfying 1.366-JR is not guaranteed to exist. We will begin by giving a reduction from clustering to TrSP which easily shows that JR is not guaranteed to exist.

Given any fair clustering instance $\mathcal{I}' = \langle \mathcal{N}', \mathcal{C}', k' \rangle$, where $|\mathcal{N}'| = n$ and $|\mathcal{C}'| = m$, we construct a corresponding TrSP instance as follows. We create two identical copies of $\mathcal{I}'$, denoted by $\mathcal{I}^a = \langle N^a, \mathcal{C}^a, k' \rangle$ and $\mathcal{I}^b = \langle N^b, \mathcal{C}^b, k' \rangle$, and place them at a sufficiently large distance from each other in a new metric space. Denote $N^a = \{a_1, a_2, \ldots, a_n\}$ and $N^b = \{b_1, b_2, \ldots, b_n\}$. We define the corresponding TrSP instance as $\mathcal{I}^T = \langle [n], \mathcal{C}^a \cup \mathcal{C}^b, (a_i, b_i)_{i \in [n]}, k = 2k' \rangle$.

**Lemma 3.4.** *Given a clustering instance $\mathcal{I}' = \langle N', \mathcal{C}', k' \rangle$, if there exists a solution $Y$ satisfying $\beta$-JR in the corresponding TrSP instance $\mathcal{I}^T$ for some $\beta \geq 1$, then there exists a solution $Y^C \subseteq Y, |Y^C| \leq k'$ such that $Y^C$ satisfies $2\beta$-PF in $\mathcal{I}'$. When $N' \cap \mathcal{C}' = \emptyset$, there exists $\varepsilon > 0$ such that $Y^C$ satisfies $(2\beta - \varepsilon)$-PF in $\mathcal{I}'$.*

*Proof.* Given a clustering instance $\mathcal{I}' = \langle N', \mathcal{C}', k' \rangle$, consider the corresponding TrSP instance $\mathcal{I}^T = \langle [n], \mathcal{C}^a \cup \mathcal{C}^b, (a_i, b_i)_{i \in [n]}, k = 2k' \rangle$ that results from the reduction described above.

Let $Y$ be a $\beta$-JR solution of $\mathcal{I}^T$ and let $Y_a = Y \cap \mathcal{C}^a$ and $Y_b = Y \cap \mathcal{C}^b$. We first observe that, since points in $\mathcal{C}^a$ and $N^a$ are an infinite distance from points in $\mathcal{C}^b$ and $N^b$, it holds for each agent $i$, that $c_i(Y) = d(a_i, Y_a) + d(b_i, Y_b)$. Moreover, as we know that $|Y| \leq 2k'$, it follows from the pigeonhole principle that either $Y_a$ or $Y_b$ has a size of at

most $k'$. Without loss of generality, we assume $|Y_a| \leq k'$.

Consider an arbitrary set of datapoints $S' \subseteq \mathcal{N}'$ with size $|S'| \geq \frac{n}{k'}$ and an arbitrary candidate center $\tau \in \mathcal{C}'$. Let $\tau_a$ and $\tau_b$ denote the copies of $\tau$ in $\mathcal{C}^a$ and $\mathcal{C}^b$, respectively. Let $S$ denote the agents in the TrSP instance $\mathcal{I}^T$ corresponding to the endpoints $S'$ and note that $|S| = |S'| \geq \frac{n}{k'} = \frac{2n}{k}$. Since $Y$ satisfies $\beta$-JR, there exists at least one agent $i \in S$ such that $\beta \cdot c_i(\{\tau_a, \tau_b\}) \geq c_i(Y)$, which implies $\beta \cdot (d(a_i, \tau_a) + d(b_i, \tau_b)) \geq d(a_i, Y_a) + d(b_i, Y_b)$. Notice that $d(a_i, \tau_a) = d(b_i, \tau_b)$ due to the construction of our reduction. Consequently, we have

$$2 \cdot \beta d(a_i, \tau_a) \geq d(a_i, Y_a) + d(b_i, Y_b) \geq d(a_i, Y_a). \quad (1)$$

It follows that for any arbitrary group of datapoints $S' \subseteq N'$ with size $|S'| \geq \frac{n}{k'}$ and candidate center $\tau \in \mathcal{C}'$, there exists a datapoint $j \in S'$ such that $2\beta \cdot d(j, \tau) \geq d(j, Y_a)$. Thus, $Y_a$ is a $2\beta$-PF solution to the original clustering instance $\mathcal{I}$.

Suppose that $\mathcal{N} \cap \mathcal{C} = \emptyset$. Let $\varepsilon = \frac{\min_{i' \in N', \tau' \in \mathcal{C}'} d(i', \tau')}{\max_{i^* \in N', \tau^* \in \mathcal{C}'} d(i^*, \tau^*)}$ be the minimum ratio between any pair of distances between agents and candidate centers in the clustering instance. Note that these ratios are well-defined and strictly positive for all agent-candidate pairs since all distances are strictly positive by our assumption. Then Equation (1) tells us that $(2\beta - \varepsilon) \cdot d(i, \tau) \geq 2\beta \cdot d(i, \tau) - d(i, Y_b) \geq d(i, Y_a)$, showing that $Y_a$ satisfies $(2\beta - \varepsilon)$-PF for $\varepsilon > 0$. $\square$

It follows easily from Lemma 3.4 that a solution exactly satisfying JR is not guaranteed to exist[7]. In fact, we are able to improve on this bound by constructing an instance for which a solution satisfying $(\frac{1+\sqrt{3}}{2} - \varepsilon)$-JR is not guaranteed to exist for any $\varepsilon > 0$ (see Section D).

**Proposition 3.5.** *For any $\varepsilon > 0$, there exists a TrSP instance for which no solution satisfies $(\frac{1+\sqrt{3}}{2} - \varepsilon)$-JR.*

The lower bound stated in Proposition 3.5 leaves open the possibility that other algorithms can significantly outperform GC-TrSP with respect to approximate JR. Recall that GC-TrSP proceeds by reducing TrSP instances to clustering instances, by reinterpreting all agents' endpoints in TrSP as datapoints in clustering, thus forfeiting information tying points to agents. Since our fairness properties ultimately consider *agent* costs, it seems likely that any clustering approach to our problem will leave significant room for improvement. The next result formalizes this intuition by showing that no clustering algorithm can achieve an approximation ratio better than 3 with respect to JR.

---

[7]To see this, assume that there exists an algorithm that always outputs a JR solution. By Lemma 3.4, this would imply the existence of an algorithm that satisfies $(2 - \varepsilon)$-PF in clustering for any instance in which $\mathcal{N} \cap \mathcal{C} = \emptyset$. However, Chen et al. (2019) give a clustering instance in which $\mathcal{N} \cap \mathcal{C} = \emptyset$ and no $(2 - \varepsilon)$-PF solution exists for any $\varepsilon > 0$, thereby yielding a contradiction.

**Proposition 3.6.** *For any $\varepsilon > 0$, there is no clustering algorithm which satisfies $(3 - \varepsilon)$-JR for the TrSP problem.*

*Proof.* Fix $\varepsilon > 0$. To prove the statement, we first define a clustering instance $\mathcal{I}^C = \langle \Theta, \mathcal{C}, k \rangle$ and then show that, no matter which solution $Y$ the clustering algorithm returns, there exists a TrSP instance $\mathcal{I}$ for which (1) $Y$ violates $(3 - \varepsilon)$-JR and (2) $\mathcal{I}^C$ is the clustering instance induced by $\mathcal{I}$. We begin by defining the clustering instance $\mathcal{I}^C$ with 12 datapoints, 9 candidate centers, and $k = 6$. Specifically, $\Theta = \{x_1, x_2, \ldots, x_{12}\}$, $\mathcal{C} = \Theta \setminus \{x_4, x_8, x_{12}\}$, where all the datapoints and centers are partitioned into three groups, each of which are separated from each other by a sufficiently large distance $H$. We represent the instance graphically by Figure 3.

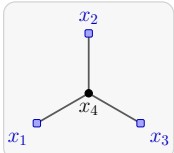 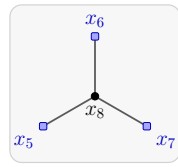 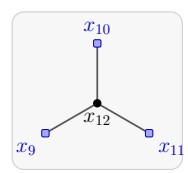

*Figure 3.* Graphical representation of clustering instance $\mathcal{I}^C$. Each edge in the graph has unit length 1 and distances between pairs of points are given by the shortest path between them (infinite distance if the pair is not connected). Datapoints which are candidate centers are labeled by blue rectangles.

Note that $k = 6$ and there are three candidate centers in each separated group. Since the internal structure in each separated group is exactly the same, we can limit our attention to two cases: (1) there is a group with zero candidate centers selected; or (2) each group has at least one selected candidate center, and there are at least two groups with at most two centers selected from each.

**Case (1).** Without loss of generality, suppose that no center is selected from the first group, i.e., suppose $Y \subseteq \{x_5, x_6, x_7, x_9, x_{10}, x_{11}\}$. Let $\mathcal{I}$ be a TrSP instance in which $\theta_1 = (x_1, x_8)$ and $\theta_2 = (x_4, x_5)$ and the remaining datapoints are arbitrarily assigned as the remaining four agents' endpoints. It is clear that $\mathcal{I}^C$ is the clustering instance induced by $\mathcal{I}$. Let $S = \{1, 2\}$. We observe that the agent group $S$ is large enough to deserve two transit stops, i.e., $|S| = 2 = \frac{2n}{k}$. Consider an alternative set of stops $T = \{x_1, x_5\}$. For each agent $i \in S$, it holds that $c_i(Y) = H$ and $c_i(T) = 1$. Therefore $Y$ gives an arbitrarily bad approximation of JR under this instance.

**Case (2).** Without loss of generality, we assume that the first two groups have at most two centers selected from each and that these centers are $Y = \{x_1, x_2, x_5, x_6\}$. Note that selecting less centers from either group could only help us in finding a deviating coalition so we are analyzing the worst case. Also, while $Y$ could contain centers from the third group, this is irrelevant to the present case

since we will not consider any endpoints in the third group when constructing our deviating coalition. Let $\mathcal{I}$ be a TrSP instance in which $\theta_1 = (x_4, x_7)$ and $\theta_2 = (x_3, x_8)$ and the remaining datapoints are arbitrarily assigned as the remaining four agents' endpoints. Again, note that $\mathcal{I}^C$ is the clustering instance induced by $\mathcal{I}$. Let $S = \{1, 2\}$ and $T = \{x_3, x_7\}$. For each agent $i \in S$, $i$ prefers stops in $T$ than $Y$ as $c_i(Y) = 2 + 1 = 3$ and $c_i(T) = 1 + 0 = 1$, which gives us $\frac{c_i(Y)}{c_i(T)} = 3 > 3 - \varepsilon$. This concludes the proof. $\square$

We close this section by remarking that clustering algorithms, besides exhibiting a JR lower bound of 3, also fail to provide any guarantee with respect to JR under instances with arbitrary transit cost functions. In the next section, we will propose an algorithm which attains a JR approximation below the lower bound stated in Proposition 3.6, and show that this holds for arbitrary transit cost functions.

*Remark* 3.7. When allowing for arbitrary transit cost functions $d'(\cdot)$, no clustering algorithm has a constant-factor approximation with respect to JR. To see this, recall the example in the proof of Proposition 3.6 and additionally define $d'(y_1, y_2) = H > 0$ and $d'(\tau_1, \tau_2) = 0$. For the deviation coalition $S$, as $d'(y_1, y_2) = H$ tends to infinity, it follows immediately that for every agent $i \in S$, we have $\frac{c_i(Y)}{c_i(T)} \to \infty$, which implies that the approximation to JR attained by GC-TrSP is unbounded under arbitrary transit cost functions.

## 4. Expanding Cost Algorithm

In the previous section, we leveraged connections with centroid clustering to show that GC-TrSP approximates JR within a $2 + \sqrt{5}$ factor. Our lower bound on approximate JR existence of $\frac{1+\sqrt{3}}{2}$ then leaves an intriguing gap. From Proposition 3.6 and the ensuing remark, we know that clustering algorithms cannot hope to improve this factor beyond 3, and furthermore, are not robust to general transit cost functions. Given the gap and the shortfalls of the clustering approach, a natural question arises: can we design algorithms that achieve better approximations to JR and are robust to non-zero transit cost functions? We answer this question affirmatively by proposing the novel *Expanding Cost Algorithm* (ECA), which fully utilizes the agent (as opposed to endpoint) information to guarantee a $1 + \sqrt{2} \approx 2.414$ approximation to JR under arbitrary transit cost functions.

ECA draws inspiration from the Greedy Capture approach of uniformly growing balls. However, instead of growing distance-based radii around individual candidate stops, it grows what we refer to as the *cost radius*, centered on *pairs* of stops. Specifically, ECA begins by enumerating all possible pairs of candidate transit stops. For each pair $T = \{\tau_1, \tau_2\}$, it uniformly expands a "cost ball", that is a set that includes all agents whose total cost when using $T$

is at most $r$.[8] The algorithm iteratively "opens" these cost balls, and adds the associated stop locations into the solution. Agents are considered active if they are not yet covered by any previously opened ball. In each iteration, the algorithm selects and opens any ball that covers at least $\lceil \frac{2n}{k} \rceil$ active agents. We formally describe ECA in Algorithm 1.

---

**Algorithm 1** Expanding Cost Algorithm

---

**Input:** TrSP instance $\mathcal{I} = \langle N, \mathcal{C}, \{\theta_i\}_{i \in N}, k \rangle$.
**Output:** Solution $Y$.
1: Initialize $r \leftarrow 0$, $Y \leftarrow \emptyset$, $\mathcal{N} \leftarrow N$.
2: **while** $\mathcal{N} \neq \emptyset$ **do**
3:     Smoothly increase $r$.
4:     **while** $\exists\, i \in N$ such that $c_i(Y) \leq r$ **do**
5:         $\mathcal{N} \leftarrow \mathcal{N} \setminus \{i\}$
6:     **end while**
7:     **while** $\exists\, \{\tau_1, \tau_2\} \subseteq \mathcal{C}, \{\tau_1, \tau_2\} \not\subseteq Y$ and $\exists\, S \subseteq \mathcal{N}, |S| \geq \lceil \frac{2 \cdot n}{k} \rceil$, such that $\forall\, j \in S, \quad c_j(Y \cup \{\tau_1, \tau_2\}) \leq r$ **do**
8:         $Y \leftarrow Y \cup \{\tau_1, \tau_2\}$
9:         $\mathcal{N} \leftarrow \mathcal{N} \setminus S$
10:     **end while**
11: **end while**
12: Return $Y$.

---

We next prove that ECA achieves a tight $(1 + \sqrt{2})$-JR guarantee, which holds for any arbitrary transit cost function.

**Theorem 4.1.** *For any arbitrary transit cost function $d'(\cdot) \geq 0$, ECA satisfies $(1 + \sqrt{2})$-JR. However, for any $\varepsilon > 0$, there exists an instance with null transit costs for which ECA violates $(1 + \sqrt{2} - \varepsilon)$-JR.*

*Proof Sketch.* Given any TrSP instance $\mathcal{I}$, let $Y \subseteq \mathcal{C}$ denote the solution by ECA. Suppose solution $Y$ violates $(1 + \sqrt{2})$-JR. It follows that there exists a group of agents $S \subseteq N$ with $|S| \geq \lceil \frac{2n}{k} \rceil$ and a pair of transit stops $T = \{\tau_1, \tau_2\}$ such that $(1 + \sqrt{2}) \cdot c_j(T) < c_j(Y)$ for all $j \in S$. We observe that every agent in $S$ uses the transit system in $T$ rather than walking and let $r_T = \max_{j \in S} c_j(T)$ denote the maximum cost for any agent in $S$ incurred under pair $T$ and agent $i^*$ denote the agent in $S$ who realizes this maximum. If the cost radius in the ECA execution never reaches $r_T$, then ECA returns stop placement $Y$ covering all the agents in $N$ with a cost radius smaller than $r_T$, which means that $c_{i^*}(Y) \leq c_{i^*}(T)$, yielding a contradiction. Conversely, if ECA does consider a cost radius of $r_T$, we proceed by proving an upper bound on the cost of $i^*$ under solution $Y$. In particular, we show that there exists another agent $i \in S$ such that

$c_i(Y) \leq r_T$ and it holds that $c_{i^*}(Y) \leq 2 \cdot r_T + c_i(T)$. By considering the minimum multiplicative cost improvement of agents $i$ and $i^*$ under $T$, we derive a contradiction by showing that $\min \left( \frac{c_i(Y)}{c_i(T)}, \frac{c_{i^*}(Y)}{c_{i^*}(T)} \right) \leq 1 + \sqrt{2}$. The detailed proof and lower bound is provided in Appendix B.2. $\qquad\square$

In light of the theoretical limitations of clustering algorithms (see Proposition 3.6 and Remark 3.7), Theorem 4.1 establishes two clear advantages of ECA over all clustering algorithms: a superior approximation to JR, and robustness of this approximation factor to arbitrary transit cost functions. To do so, ECA explicitly considers pairs of stops at a time, and in this way, assigns agents to routes as it goes, rather than simply assigning endpoints to stops. While this approach outperforms clustering algorithms in the sense of satisfying coalitions who all desire the same pair of stops, those guarantees do not extend to coalitions who prefer to deviate to larger sets of stops. Indeed, as we will now show, the approach of ECA is too myopic to guarantee any bounded approximation to the core.

**Proposition 4.2.** *For any $\gamma, \rho \geq 1$, there exists a TrSP instance in which ECA fails $(\gamma, \rho)$-core.*

Intuitively, we construct an instance based on the complete graph $K_z$, where each vertex corresponds to a candidate stop, and each edge contains two additional internal candidate stops and $r$ agents. By setting the size $k = z(z - 1)$, ECA's capture threshold is exactly $\frac{2n}{k} = r$. For each edge, the two internal edge-stops capture exactly the $r$ agents on that edge, so ECA selects all internal edge-stops and exhausts its entire budget. However, the vertex-agents across all edges together form a sufficiently large coalition that can deviate to the $z$ vertex stops, reducing their cost from 2 to 0, thereby violating any $(\gamma, \rho)$-approximation of core. Despite guaranteeing the best-known approximation to JR, ECA performs arbitrarily poorly with respect to core, while on the other hand, GC-TrSP obtains a worse JR approximation but guarantees a constant-factor core approximation. So we present a parameterized algorithm, termed $\lambda$-Hybrid algorithm, which effectively navigates the tradeoff between JR and core delineated by ECA and GC-TrSP in Section E.2.

## 5. Experiments

We complement our theoretical contributions by evaluating the empirical performance of the GC-TrSP algorithm, the Expanding Cost algorithm, and the $\lambda$-Hybrid algorithm (Appendix E) on a real world dataset. All source code is provided in the supplementary material.

**Experimental Setup** We use resident travel route data from the City of Helena Capital Transit service, comprising 10,282 distinct travel routes between 3,075 unique spatial

---

[8]We note that what we refer to as cost balls are not in fact balls in the geometric sense. We use the ball terminology nevertheless as it lends a natural interpretation to ECA, and especially the algorithm we introduce in Section E.

points. The initial dataset specifies only pick-up and drop-off locations. Using OpenStreetMap data (OpenStreetMap contributors, 2026) and the open-source *Valhalla* routing engine (Valhalla contributors, 2026), we compute route-level travel times for both walking and public transit, which serve as the corresponding cost metrics. For the JR and core experiments, we randomly sample 400 and 40 agents[9], respectively, together with their associated routes from the full dataset. We define the candidate stop set as the union of all observed locations among the sampled agents. For the target number of stops, we evaluate JR over $k$ in $[20, 100]$ and core over $k$ in $[5, 20]$. Finally, because the Helena dataset exhibits substantial disparities between walking and transit costs, we additionally rescale transit costs over a broad range to assess algorithmic performance under varying degrees of separation between transit and walking scales. We sample 50 rounds for each parameter combination.

**JR Evaluation.** We evaluate the approximation performance with respect to JR by simulating GC-TrSP and ECA across a range of stop selection sizes and transit cost scales.

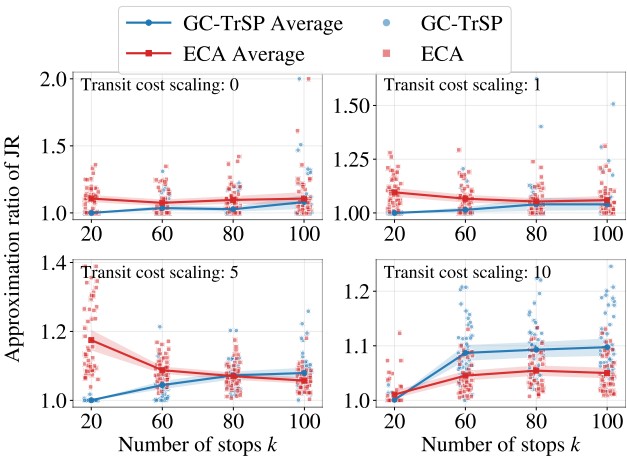

*Figure 4.* JR approximation evaluation with 400 agents. Stop selection size ranges from 20 to 100, and the transit cost scale ranges from 0 to 10. Distribution of instance approximation ratios and the mean approximation ratio with 95% confidence intervals

In Figure 4, we first observe that for both GC-TrSP and ECA, the approximation ratio of JR under random sampling is close to 1, suggesting that our theoretical lower bounds may not be borne out in practice. Moreover, despite our result that ECA admits a stronger worst case bound for JR, GC-TrSP delivers approximation ratios that remain close to one across all stop selection sizes. ECA exhibits slightly larger ratios and greater variability when the transit costs remain low, with the most pronounced separation around

---

[9]The core test uses a small sample size because verifying core membership is computationally intensive and requires solving large scale integer programs.

$k = 20$, followed by improvement as $k$ increases. The trend shifts when the transit cost is scaled by a larger factor, representing scenarios where walking (or other modes) may be a feasible alternative for a significant number of routes. In this regime, the performance of GC-TrSP deteriorates as $k$ grows, whereas ECA is stable and becomes competitive, achieving better approximation ratios for larger sizes.

**Core Evaluation**. We fix the size relaxation parameter ($\alpha$) to 2 and evaluate the cost approximation ratios ($\beta$) of both algorithms and exhibit the performance of GC-TrSP and ECA under the zero transit cost (transit cost scaling $= 0$) and regular transit cost settings (transit cost scaling $= 1$) in Table 2. More results under other transit cost scalings can be found in the Appendix F. Across all stop selection sizes and transit cost scales, GC-TrSP attains better core approximation ratios than ECA. While ECA displays higher means and greater variability in several regimes, its approximation ratios remain close to 1 for most instances, which indicates that ECA is still practically effective in our experiments, despite admitting an arbitrarily poor worst-case guarantee.

| Zero Transit Cost (Scaling $= 0$) | | | | | | | | |
|---|---|---|---|---|---|---|---|---|
| **Results** | $k = 5$ | | $k = 10$ | | $k = 15$ | | $k = 20$ | |
| | GC | ECA | GC | ECA | GC | ECA | GC | ECA |
| **Average** | 1.016 | 1.114 | 1.095 | 1.430 | 1.155 | 1.365 | 1.074 | 1.269 |
| **Min** | 1.000 | 1.000 | 1.000 | 1.074 | 1.000 | 1.081 | 1.000 | 1.005 |
| **Max** | 1.263 | 1.301 | 1.556 | 2.095 | 1.533 | 1.743 | 1.241 | 1.675 |

| Regular Transit Cost (Scaling $= 1$) | | | | | | | | |
|---|---|---|---|---|---|---|---|---|
| **Results** | $k = 5$ | | $k = 10$ | | $k = 15$ | | $k = 20$ | |
| | GC | ECA | GC | ECA | GC | ECA | GC | ECA |
| **Average** | 1.007 | 1.072 | 1.053 | 1.454 | 1.067 | 1.386 | 1.058 | 1.190 |
| **Min** | 1.000 | 1.000 | 1.000 | 1.138 | 1.000 | 1.041 | 1.000 | 1.000 |
| **Max** | 1.082 | 1.256 | 1.240 | 1.889 | 1.241 | 1.787 | 1.257 | 1.499 |

*Table 2.* Core approximation evaluation with 40 agents under zero and regular transit costs. We consider both zero and regular transit costs. GC denotes GC-TrSP. For each setting, we report the average, minimum, and maximum values over 50 sampled rounds.

## 6. Discussion

Our central algorithmic result shows that ECA satisfies 2.414-JR under arbitrary transit cost metrics assumption. Our core approximation upper bounds, by contrast, rely on a connection to proportional fairness in clustering and therefore hold only under null transit costs. Whether constant-factor approximations to the core can be achieved under broader classes of transit cost metrics remains open. In one sense, TrSP problem can be viewed as a generalization of centroid selection in which each agent has two points rather than one. For future work, a natural one is to allow each agent to have multiple points that must be traversed. The first challenge of the extension would be defining a sensible cost function for agents. It would then be interesting to see how well the results of this paper extend to such a setting.

## Acknowledgements

The authors thank Vinayak Dixit and Taha Hossein Rashidi for valuable comments, and the City of Helena for facilitating access to the records used in our experimental evaluation. The UNSW authors are supported by the NSF-CSIRO grant on "Fair Sequential Collective Decision-Making" (RG230833). A portion of this work was completed while Jeremy Vollen was a student at UNSW.

## Impact Statement

This paper studies algorithmic approaches to fair transit stop placement, with the goal of helping planners make more transparent and equitable decisions when allocating limited mobility infrastructure. Such methods could better represent groups of riders with similar travel needs, especially in shuttle, community transit, or fixed-route settings.

The model focuses on stop placement and treats transit times between stops as exogenous, so it is not intended for full-scale public transportation network design where route selection substantially affects riders' costs. It is better suited to settings where routes are fixed in advance or where walking/access time dominates total travel time. In deployment, fairness guarantees also depend on representative travel-demand data and do not capture all relevant concerns, such as accessibility, privacy, safety, environmental justice, or the needs of small vulnerable groups. Thus, these algorithms should be used as supporting tools alongside privacy-preserving data practices, robustness checks, and community input for decisions.

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

## A. Further Related Work

The problem of transit stop placement has been extensively studied in the public transportation literature (Ceder & Wilson, 1986; Ceder, 2016; Chien* & Qin, 2004). For instance, Hossein Rashidi et al. (2016) investigated the optimization of stop locations in transit networks under elastic travel demand and budget constraints. They formulated the problem as a mixed integer program and developed heuristic algorithms capable of solving large-scale instances. Ceder et al. (2015) addressed the challenge of bus stop placement in routes with uneven topography. They incorporated topographical variation into three distinct mathematical models that account for walking speed, the attractiveness of access paths to transit services, and vehicle acceleration at stops. To tackle it, they proposed a heuristic evolutionary algorithm to approximate optimal solutions.

Fair transit stop placement has received growing attention in recent years due to its critical role in promoting equitable access to public transportation (Martens, 2016; Martens & Lucas, 2018). For instance, Tedjopurnomo et al. (2022) studied the equitable public bus network optimization problem through a case study of Singapore's bus system, formulating efficiency and equity metrics and conducting exploratory experiments to evaluate their real-world impact. Their work underscores the challenges of balancing fairness and efficiency in transit network design. In a complementary direction, Najmi et al. (2023) explored fairness and equity from psychological and cognitive perspectives, highlighting how users' perceptions and experiences shape their sense of fairness in public transit. Matl et al. (2018) wrote a comprehensive survey on equitable vehicle routing problems (VRPs), reviewing various strategies to embed fairness into routing decisions. More recently, He et al. (2024) examined fairness in transportation network design from a welfarist perspective, focusing on selecting a subset of edges in an undirected graph to optimize network performance while maintaining fairness considerations.

In this paper, we consider fairness notions inspired by the concept of *core stability*, a foundational idea that has been extensively applied across various domains, including transferable utility cooperative games (Peleg & Sudhölter, 2007), coalition formation (Aziz & Savani, 2016), exchange markets (Shapley & Scarf, 1974), and two-sided matching (Alkan & Gale, 1990). Although it is typically viewed as a stability concept, it is also amongst the strongest fairness notions in many social choice contexts, including committee voting (Lackner & Skowron, 2023), fair mixing (Aziz et al., 2020), and fair allocation (Fain et al., 2018).

Bullinger et al. (2025) were the first to explore fairness in the transit stop placement problem. Their work focused on the line metric. They presented polynomial-time algorithms for cost minimization and designed algorithms with provable fairness guarantees, including justified representation (JR) and a factor-two approximation for core stability. Importantly, they note that "an important research challenge is to develop a richer framework that can be used to reason about fairness in more realistic models of public transport." Motivated by this, we consider a more general setting in which agents and candidate bus stops lie in an arbitrary metric space. In contrast to the positive results Bullinger et al. (2025) obtained for the line, we will show that in general, the cost minimization problem becomes computationally intractable, and a solution satisfying JR need not exist.

The TrSP problem we study in this paper is closely related to the study of fairness in centroid selection for clustering (Chhabra et al., 2021; Chen et al., 2019; Micha & Shah, 2020; Aziz et al., 2024; Kellerhals & Peters, 2024). Chen et al. (2019) initiated the study of fairness in clustering, introducing the proportional fairness (PF) axiom, requiring that no "large enough"group of datapoints has an incentive to collectively deviate to an unselected candidate center, an idea inspired by core stability. They showed that outcomes satisfying reasonable approximations of PF are guaranteed to exist and can be computed via the "Greedy Capture" algorithm. Following this, a substantial body of work has emerged on fairness and proportionality in clustering. For example, Micha & Shah (2020) extended the analysis to unconstrained centroid candidate sets; Aziz et al. (2024) proposed a fairness axiom targeting proportional representation; and Kellerhals & Peters (2024) established connections between fair clustering and committee voting, and analyzed the connections between several of the axioms introduced in this literature. The TrSP problem studied in this paper can be viewed as a variant of centroid selection, where centroids correspond to transit stops. However, unlike standard clustering where each datapoint represents an agent, the TrSP model associates two datapoints with each agent. In this work, we will highlight both the inherent connections and the fundamental differences between these two problem domains.

## B. Minimizing Total Travel Cost

**Proposition 2.1.** *Unless $P = NP$, there is no polynomial time algorithm which computes a minimum cost solution to the TrSP problem, even under null transit cost, i.e., even when $d'(i, j) = 0$, for all $i, j \in \mathcal{X}$.*

*Proof.* We prove the statement by a reduction from the canonical $k$-median problem in general metric space, which is known to be NP-hard (Megiddo & Supowit, 1984).

**$k$-median decision problem.** Given a metric space $(\mathcal{X}, d)$, a set $N$ of datapoints $\{x_i\}_{i \in N}$, a set $\mathcal{C}'$ of candidate centers, an integer $k'$, and a bound $B$, the decision problem asks whether there exists a set $Y \subseteq \mathcal{C}'$ with $|Y| \leq k'$ such that

$$\sum_{i \in N} d(x_i, Y) \leq B.$$

**TrSP decision problem (with null transit times).** Let $\mathcal{I} = \langle N, \mathcal{C}, k, \theta_i i \in N \rangle$ be a TrSP instance with null transit times in metric space $(\mathcal{X}, d)$. The $\tau$-TrSP decision problem asks whether there exists a set $Y \subseteq \mathcal{C}$ with $|Y| \leq k$ such that

$$\sum_{i \in N} c_i(Y) \leq \tau.$$

Given an arbitrary $k$-median instance in metric space $(\mathcal{X}, d)$, we construct a corresponding TrSP instance as follows. We augment the metric space by adding additional points, $x'$ and $c'$, which are both located at a distance $B$ from every other point in the metric space and distance 0 from each other, i.e., $\forall x \in \mathcal{X} \setminus \{x', c'\}, d(x', x) = d(c', x) = B$ and $d(x', c') = 0$. We then let $\theta_i = (x_i, x')$ be the endpoints of each agent $i \in N$, and let $\mathcal{C} = \mathcal{C}' \cup \{c'\}$. Lastly, we set the number of desired transit stops to $k = k' + 1$ and let $\tau = B$.

Suppose the $k$-median instance is a YES instance, i.e., there exists a solution $Y \subseteq \mathcal{C}'$ with $|Y| \leq k' = k - 1$ such that $\sum_{i \in N} d(x_i, Y) \leq B$. We show that $Y^* = Y \cup \{c'\}$ is a YES solution for the constructed TrSP instance:

$$\begin{aligned}
\sum_{i \in N} c_i(Y^*) &= \sum_{i \in N} \min\{d(x_i, x'), \min_{y_1, y_2 \in Y^*} d(x_i, y_1) + d(x', y_2)\} \\
&= \sum_{i \in N} \min\{B, \min_{y_1 \in Y} d(x_i, y)\} && (c' \in Y^*, d(x', c') = 0) \\
&\leq \sum_{i \in N} d(x_i, Y) \\
&\leq B.
\end{aligned}$$

Conversely, suppose the TrSP instance is a YES instance, i.e., there exists $Y$ with $|Y| \leq k$ such that $\sum_{i \in N} c_i(Y) \leq B$. First observe that $c' \in Y$, since otherwise every agent's cost will be at least $B$. Also note that $\min_{y \in Y} d(x_i, y) \leq B$ for every $i \in N$. Now consider the set $Y^* = Y \setminus \{c'\}$. Since $|Y^*| = |Y| - 1 \leq k - 1 = k'$, we know that $Y^*$ is a feasible solution to the $k$-median instance. We see that $Y^*$ is a certificate that the $k$-median instance is a YES instance by noting that

$$\begin{aligned}
\sum_{i \in N} d(x_i, Y^*) &= \sum_{i \in N} \min_{y \in Y^*} d(x_i, y) \\
&= \sum_{i \in N} \min\{d(x_i, x'), \min_{y \in Y^*} d(x_i, y)\} && (\min_{y \in Y} d(x_i, y) \leq B) \\
&= \sum_{i \in N} \min\{d(x_i, x'), \min_{y_1, y_2 \in Y} d(x_i, y_1) + d(x', y_2)\} \\
&= \sum_{i \in N} c_i(Y) \\
&\leq B.
\end{aligned}$$

The third transition follows because $c' \in Y$ and thus choosing $y_2$ to be $c'$ incurs no additional cost. Since the above reduction is polynomial time, the TrSP decision problem is NP-hard. $\square$

## C. Omitted Pseudocode

We present formal pseudocode for the Greedy Capture algorithm in the context of the Transit Stop Selection (TrSP) problem, referred to as the GC-TrSP algorithm in Algorithm 2.

---

**Algorithm 2** Greedy Capture for TrSP (GC-TrSP)

---

**Input:** TrSP instance $\mathcal{I} = \langle N, \mathcal{C}, \{\theta_i\}_{i \in N}, k \rangle$.
**Output:** Solution $Y$.
 1: Create a clustering instance $\mathcal{I}^C = \langle \Theta, \mathcal{C}, k \rangle$ induced by $\mathcal{I}$.
 2: Denote the distance ball of candidate $c \in \mathcal{C}$ with radius $r$ by $B(c, r) \leftarrow \{j \in \Theta : d(j, c) \leq r\}$.
 3: Initialize $r \leftarrow 0$, $Y \leftarrow \emptyset$; $\mathcal{N} \leftarrow \Theta$.
 4: **while** $\mathcal{N} \neq \emptyset$ **do**
 5:     Smoothly increase $r$.
 6:     **while** $\exists\, y \in Y$ s.t. $|B(y, r) \cap \mathcal{N}| \geq 1$ **do**
 7:         $\mathcal{N} \leftarrow \mathcal{N} \setminus B(y, r)$
 8:     **end while**
 9:     **while** $\exists\, c \in \mathcal{C} \setminus Y$ s.t. $|B(c, r) \cap \mathcal{N}| \geq \lceil \frac{2n}{k} \rceil$ **do**
10:         $Y \leftarrow Y \cup \{c\}$
11:         $\mathcal{N} \leftarrow \mathcal{N} \setminus B(c, r)$
12:     **end while**
13: **end while**
14: Return $Y$.

---

## D. Omitted Proofs

Some of the proofs contained herein correspond only to the lower bound portion of the statements. In these cases, the proof of the upper bound portion can be found in the main body of the paper.

**Proposition 3.2.** *GC-TrSP satisfies $(2, 1 + \sqrt{2})$-core. However, for any $\delta, \varepsilon > 0$, there exists an instance for which GC-TrSP violates $(2 - \delta, 1 + \sqrt{2} - \varepsilon)$-core.*

*Proof.* Since Chen et al. (2019) has shown that the Greedy Capture algorithm achieves a $(1 + \sqrt{2})$-PF guarantee in general fair clustering, it follows that for any TrSP instance $\mathcal{I}$, the GC-TrSP algorithm satisfies $(1 + \sqrt{2})$-PF in the induced clustering instance $\mathcal{I}^C$. By Theorem 3.1, this implies that GC-TrSP satisfies the $(2, 1 + \sqrt{2})$-core.

To prove the bound is tight, let $H = \lceil 1/(3 \cdot \delta) \rceil$. We construct an instance with $n = 15 \cdot H$ agents $N$, $m = 7$ candidate transit stops $\mathcal{C} = \{\tau_1, \tau_2, \ldots, \tau_7\}$, and $k = 5$. The agent set $N$ is partitioned into three subsets, each consisting of agents located at distinct but internally identical locations. Specifically, we define $N = N_1 \cup N_2 \cup N_3$, where $|N_1| = |N_2| = 6 \cdot H - 1$, and $|N_3| = 3 \cdot H + 2$. For each group $N_i$, all agents share the same travel locations, denoted $(a_i, b_i)$. The distance from transit stops $\{\tau_1, \ldots, \tau_4\}$ to locations $\{a_1, b_1, a_2, b_2, a_3, b_3\}$ are specified in Table 3. For the remaining stops $\{\tau_5, \tau_6, \tau_7\}$, we assign a large constant distance to each endpoint in $\{a_1, b_1, a_2, b_2, a_3, b_3\}$.

| $d(\cdot)$ | $a_1$ | $a_2$ | $a_3$ | $b_1$ | $b_2$ | $b_3$ |
|---|---|---|---|---|---|---|
| $\tau_1$ | $1$ | $\sqrt{2} - 1$ | $1$ | $\infty$ | $\infty$ | $\infty$ |
| $\tau_2$ | $1 + \sqrt{2} - \varepsilon$ | $1 - (\sqrt{2} - 1)\varepsilon$ | $1 - (\sqrt{2} - 1)\varepsilon$ | $\infty$ | $\infty$ | $\infty$ |
| $\tau_3$ | $\infty$ | $\infty$ | $\infty$ | $1$ | $\sqrt{2} - 1$ | $1$ |
| $\tau_4$ | $\infty$ | $\infty$ | $\infty$ | $1 + \sqrt{2} - \varepsilon$ | $1 - (\sqrt{2} - 1)\varepsilon$ | $1 - (\sqrt{2} - 1)\varepsilon$ |

*Table 3.* An instance in which GC-TrSP fails $(2 - \delta, 1 + \sqrt{2} - \varepsilon)$-core.

Keeping in mind that $\lceil \frac{2n}{k} \rceil = 6 \cdot H$, we describe the execution of GC-TrSP on this instance. The minimum radius ball that captures $6 \cdot H$ endpoints is centered at $\tau_2$ (and $\tau_4$) with radius $1 - (\sqrt{2} - 1)\varepsilon$. Thus, GC-TrSP selects $\{\tau_2, \tau_4\}$, and deactivates all of the endpoints of the agents in $N_2 \cup N_3$. After that, there is no candidate which can capture $6 \cdot H$ endpoints with a distance radius less than $\infty$, so endpoints located at $a_1$ ($b_1$) are deactivated by $\tau_2$ ($\tau_4$). As a result, GC-TrSP returns solution $Y = \{\tau_2, \tau_4\}$.

Now consider the set of agents $S = N_1 \cup N_2$ and candidate stop set $T = \{\tau_1, \tau_3\}$. For each agent $i \in N_1$, it holds that $\frac{c_i(Y)}{c_i(T)} = \frac{2(1 + \sqrt{2} - \varepsilon)}{2} = 1 + \sqrt{2} - \varepsilon$. For each agent $i \in N_2$, it holds that $\frac{c_i(Y)}{c_i(T)} = \frac{2(1 - (\sqrt{2} - 1)\varepsilon)}{2(\sqrt{2} - 1)} = 1 + \sqrt{2} - \varepsilon$. Furthermore,

we have

$$|S| = 12 \cdot H - 2 \geq (2 - \frac{1}{3 \cdot H}) \cdot \frac{2n}{k} \geq (2 - \delta) \cdot \frac{|T| \cdot n}{k}.$$

It follows that there exists such a blocking coalition $S$ and a candidate stop subset $T$ with size $|S| \geq (2 - \delta) \cdot \frac{|T| \cdot n}{k}$ such that for every agent $i \in S$, we have $(1 + \sqrt{2} - \varepsilon) \cdot c_i(T) < c_i(Y)$. This implies that the GC-TrSP algorithm violates $(2 - \delta, 1 + \sqrt{2} - \varepsilon)$-core. □

**Theorem 3.3.** *GC-TrSP satisfies $(2 + \sqrt{5})$-JR. However, for any $\varepsilon > 0$, there exists an instance for which GC-TrSP violates $(2 + \sqrt{5} - \varepsilon)$-JR.*

*Proof.* Given any TrSP instance $\mathcal{I}$, let $Y \subseteq \mathcal{C}$ be the solution of GC-TrSP under $\mathcal{I}$. Assume, for the sake of a contradiction, that $Y$ fails to satisfy $(2 + \sqrt{5})$-JR. That is, there exists a group of agents $S \subseteq N$ with $|S| \geq \lceil \frac{2n}{k} \rceil$ and a pair of transit stops $T = \{\tau_1, \tau_2\} \subseteq \mathcal{C}$ such that $(2 + \sqrt{5}) \cdot c_j(T) < c_j(Y)$ for every agent $j \in S$.

Without loss of generality, we assume that every agent $i \in S$ travels from transit stop $\tau_1$ to $\tau_2$, walking from their starting point $a_i$ to $\tau_1$ and from $\tau_2$ to their destination $b_i$. Define

$$r_T = \max_{j \in S}\{\max\{d(a_j, \tau_1), d(b_j, \tau_2)\}\}$$

as the maximum distance between any endpoint of an agent in $S$ and two transit stops in $\{\tau_1, \tau_2\}$. Let $i^* \in S$ be the agent that attains this maximum distance. Without loss of generality, assume that this maximum distance is realized at the starting point $a_{i^*}$, i.e, $r_T = d(a_{i^*}, \tau_1)$.

We begin by considering the case in which the distance radius explored by GC-TrSP never reaches $r_T$. That is, GC-TrSP deactivates all agent endpoints, and in particular both of the endpoints of agent $i^*$, with a distance radius at most $r_T$. In this scenario, we derive that $c_{i^*}(Y) \leq 2 \cdot r_T \leq 2 \cdot c_{i^*}(T)$, which contradicts the assumption that $(2 + \sqrt{5}) \cdot c_{i^*}(T) < c_{i^*}(Y)$.

Henceforth, we focus on the case in which GC-TrSP does consider a distance radius of $r_T$ during its execution. We begin by examining the subcase where $Y$ contains one of the stops in $T$. Without loss of generality, we assume that $\tau_1 \in Y$ and $\tau_2 \notin Y$. Since $\tau_1$ is included in $Y$, all starting points $a_i$ of agents in $S$ are deactivated when the distance radius reaches at most $r_T$. Note that for stop $\tau_2$, if selected, could deactivate $\lceil \frac{2n}{k} \rceil$ endpoints with radius $r_T$. However, since $\tau_2 \notin Y$, there must exist some agent $i'$ whose endpoint $b_{i'}$ is already deactivated by another transit stop, denoted $y \in Y$, with a radius at most $r_T$. Consequently, we have $c_{i'}(Y) \leq d(a_{i'}, \tau_1) + d(b_{i'}, y) \leq d(a_{i'}, \tau_1) + r_T$. With this inequality in hand, we now proceed to establish an upper bound incurred by agent $i^*$ under the solution $Y$.

$$
\begin{aligned}
c_{i^*}(Y) &= \min_{y_1, y_2 \in Y}\{d(a_{i^*}, y_1) + d(b_{i^*}, y_2)\} \\
&\leq d(a_{i^*}, \tau_1) + d(b_{i^*}, y) && (\because \tau_1 \in Y, y \in Y) \\
&\leq r_T + d(b_{i^*}, \tau_2) + d(\tau_2, b_{i'}) + d(b_{i'}, y) && (\because \text{triangle inequality}) \\
&\leq 3 \cdot r_T + d(b_{i'}, \tau_2). && (\because d(b_{i^*}, \tau_2) \leq r_T, d(b_{i'}, y) \leq r_T)
\end{aligned}
$$

We next consider the minimum multiplicative cost improvement of agent $i'$ and $i^*$ under $T$:

$$
\begin{aligned}
\min\left(\frac{c_{i'}(Y)}{c_{i'}(T)}, \frac{c_{i^*}(Y)}{c_{i^*}(T)}\right) &\leq \min\left(\frac{d(a_{i'}, \tau_1) + r_T}{d(a_{i'}, \tau_1) + d(b_{i'}, \tau_2)}, \frac{3 \cdot r_T + d(b_{i'}, \tau_2)}{r_T}\right) \\
&\leq \min\left(\frac{r_T}{d(b_{i'}, \tau_2)}, \frac{3 \cdot r_T + d(b_{i'}, \tau_2)}{r_T}\right) \\
&\leq \max_{z \geq 1}(\min(z, 3 + 1/z)) = \frac{3 + \sqrt{13}}{2}.
\end{aligned}
$$

The second inequality holds because $d(b_{i'}, \tau_2) \leq r_T$ and subtracting $d(a_{i'}, \tau_1)$ from numerator and denominator weakly increases the resulting fraction. Therefore, we have $\frac{3 + \sqrt{13}}{2} \cdot c_{i^*}(T) \geq c_{i^*}(Y)$, which contradicts that $(2 + \sqrt{5}) \cdot c_j(T) < c_j(Y)$ for every agent $j \in S$.

The remaining subcase is when neither $\tau_1$ nor $\tau_2$ are included in $Y$. In this case, since $\tau_1$ and $\tau_2$ are excluded from $Y$, we observe that there exists at least one agent $i' \in S$ such that $a_{i'}$ is deactivated by some $y' \in Y$ with a radius at most $r_T$ and one agent $i'' \in S$ such that $b_{i''}$ is deactivated by some $y'' \in Y$ with a radius at most $r_T$. We next upper-bound the cost incurred by agents $i^*, i'$, and $i''$ under solution $Y$.

For agent $i^*$, we have

$$
\begin{aligned}
c_{i^*}(Y) &\le d(a_{i^*}, y') + d(b_{i^*}, y'') \\
&\le d(a_{i^*}, \tau_1) + d(\tau_1, a_{i'}) + d(a_{i'}, y') + d(b_{i^*}, \tau_2) + d(\tau_2, b_{i''}) + d(b_{i''}, y'') \\
&\le 3 \cdot r_T + d(b_{i^*}, \tau_2) + d(\tau_1, a_{i'}) + d(\tau_2, b_{i''}).
\end{aligned}
$$

For agent $i'$, we have

$$
\begin{aligned}
c_{i'}(Y) &\le d(a_{i'}, y') + d(b_{i'}, y'') \\
&\le d(a_{i'}, y') + d(b_{i'}, \tau_2) + d(\tau_2, b_{i''}) + d(b_{i''}, y'') \\
&\le 2 \cdot r_T + d(b_{i'}, \tau_2) + d(\tau_2, b_{i''}).
\end{aligned}
$$

We obtain the upper bound for agent $i''$ in an analogous fashion to the previous bound:

$$
c_{i''}(Y) \le 2 \cdot r_T + d(a_{i''}, \tau_1) + d(\tau_1, a_{i'}).
$$

For ease of expression, we denote $x = d(a_{i'}, \tau_1)$ and $y = d(\tau_2, b_{i''})$. With the upper bounds in hand, we consider the minimum multiplicative cost improvement of agents $\{i^*, i', i''\}$ as follows.

$$
\begin{aligned}
\min\left( \frac{c_{i'}(Y)}{c_{i'}(T)}, \frac{c_{i''}(Y)}{c_{i''}(T)}, \frac{c_{i^*}(Y)}{c_{i^*}(T)} \right) &\le \min\left( \frac{2 \cdot r_T + d(b_{i'}, \tau_2) + y}{x + d(b_{i'}, \tau_2)}, \frac{2 \cdot r_T + d(a_{i''}, \tau_1) + x}{d(a_{i''}, \tau_1) + y}, \frac{3 \cdot r_T + d(b_{i^*}, \tau_2) + x + y}{r_T + d(b_{i^*}, \tau_2)} \right) \\
&\le \min\left( \frac{2 \cdot r_T + y}{x}, \frac{2 \cdot r_T + x}{y}, \frac{3 \cdot r_T + x + y}{r_T} \right).
\end{aligned}
$$

We next consider the upper bound of the term by

$$
\min\left( \frac{2 \cdot r_T + y}{x}, \frac{2 \cdot r_T + x}{y}, \frac{3 \cdot r_T + x + y}{r_T} \right) \le \min\left( \frac{2r_T + x}{x}, 3 + \frac{2x}{r_T} \right) \le \max_{q \ge 1}\left( \min\left( 2q + 1, 3 + \frac{2}{q} \right) \right),
$$

where we can assume that $x = y$, yielding the first inequality. The reason is that if $x = 0$ or $y = 0$, our inequality gives a better bound than $2 + \sqrt{5}$. So we restrict to $x, y > 0$. We claim that $\frac{2r_T + x}{x} \ge \min(\frac{2r_T + y}{x}, \frac{2r_T + x}{y})$ for any $x, y > 0$. To see it, we divide into two cases. (1). When $y \le x$, since $x, y > 0$, we have $\frac{2r_T + x}{x} - \frac{2r_T + y}{x} = \frac{x - y}{x} \ge 0$; (2). When $y > x$, we immediately have $\frac{2r_T + x}{x} \ge \frac{2r_T + x}{y} \ge \min(\frac{2r_T + y}{x}, \frac{2r_T + x}{y})$. For the second inequality, it follows by setting $q = \frac{r_T}{x} \ge 1$ as $r_T \ge x$ by definition. Hence, we upper-bound the term by $\max_{q \ge 1}\left( \min\left( 2q + 1, 3 + \frac{2}{q} \right) \right) \le 2 + \sqrt{5}$, which implies that $\min\left( \frac{c_{i'}(Y)}{c_{i'}(T)}, \frac{c_{i''}(Y)}{c_{i''}(T)}, \frac{c_{i^*}(Y)}{c_{i^*}(T)} \right) \le 2 + \sqrt{5}$, contradicting that $(2 + \sqrt{5}) \cdot c_j(T) < c_j(Y)$ for every agent $j \in S$. So we conclude that GC-TrSP (Algorithm 2) satisfies $(2 + \sqrt{5})$-JR.

To show the tightness of the analysis, we provide the following instance in which the Greedy Capture for TrSP (Algorithm 2) fails to achieve $(2 + \sqrt{5} - \varepsilon)$-JR. Consider the TrSP instance pictured in Figure 5.

We consider $n = 7$ agents, whose endpoints are represented by blue circles. There are 6 candidate transit stops, marked as red squares, and $k = 4$. The distances are specified in the figure, and can be considered as two lines separate by an infinite distance, i.e., the distance between any pair of points lying on the same line is the sum of the distances of the intervals between them.. We note that $\lceil \frac{2n}{k} \rceil = \lceil \frac{14}{4} \rceil = 4$. It follows that, once a candidate stop can deactivate 4 active endpoints, it is selected by GC-TrSP and the corresponding endpoints are deactivated. We observe that $y_1$ and $y_2$ are the first two candidate stops selected by GC-TrSP. Afterward, no further candidate stop is selected, as no candidate stop can deactivate at least 4 active endpoints within a distance radius of at most $2 + \frac{\sqrt{5} - 1}{2} - \frac{\varepsilon}{8}$. Once the radius reaches $2 + \frac{\sqrt{5} - 1}{2} - \frac{\varepsilon}{8}$, all the endpoints

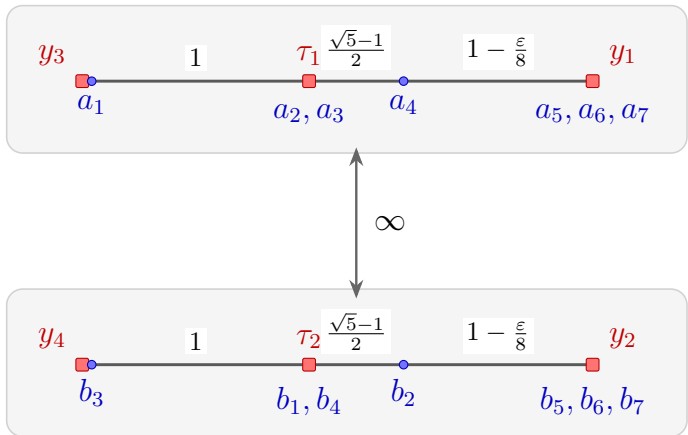

*Figure 5.* An instance where GC-TrSP algorithm fails $(2 + \sqrt{5} - \varepsilon)$-JR for any $\varepsilon > 0$.

are deactivated by the selected stops $Y = \{y_1, y_2\}$, which forms the final solution produced by the GC-TrSP algorithm. Under this solution $Y = \{y_1, y_2\}$, the costs incurred by agents $\{1, 2, 3, 4\}$ are computed as follows:

$$c_1(Y) = c_3(Y) = 2 + \sqrt{5} - \frac{\varepsilon}{4}; \; c_2(Y) = c_4(Y) = \frac{3 + \sqrt{5}}{2} - \frac{\varepsilon}{4}.$$

Notice that $S = \{1, 2, 3, 4\}$ forms a deviation coalition that prefers the alternative stop pair $T = \{\tau_1, \tau_2\}$. Under solution $T$, the costs are

$$c_1(T) = c_3(T) = 1, c_2(T) = c_4(T) = \frac{\sqrt{5} - 1}{2}.$$

Thus, for each agent $i \in \{1, 3\}$, we have $\frac{c_i(Y)}{c_i(T)} = 2 + \sqrt{5} - \frac{\varepsilon}{4} > 2 + \sqrt{5} - \varepsilon$ and for agent $i \in \{2, 4\}$, we have

$$\frac{c_i(Y)}{c_i(T)} = \frac{\frac{3 + \sqrt{5}}{2} - \frac{\varepsilon}{4}}{\frac{\sqrt{5} - 1}{2}} = 2 + \sqrt{5} - \frac{(\sqrt{5} + 1)\varepsilon}{8} > 2 + \sqrt{5} - \varepsilon.$$

Therefore, we derive that for any agent $i \in S$, $(2 + \sqrt{5} - \varepsilon) \cdot c_i(T) < c_i(Y)$, implying that solution $Y$ by GC-TrSP violates $(2 + \sqrt{5} - \varepsilon)$-JR. $\qquad\square$

**Proposition 3.5.** *For any $\varepsilon > 0$, there exists a TrSP instance for which no solution satisfies $(\frac{1 + \sqrt{3}}{2} - \varepsilon)$-JR.*

*Proof.* Consider a TrSP instance with 3 agents, 6 transit candidate stops, and $k = 3$. Distances are specified in the following table. We first observe that the endpoints and stops respect the triangle inequality and are partitioned into two distinct

| $d(\cdot)$ | $a_1$ | $a_2$ | $a_3$ | $b_1$ | $b_2$ | $b_3$ |
|---|---|---|---|---|---|---|
| $\tau_1$ | $2 + \sqrt{3}$ | $\sqrt{3}$ | $1$ | $\infty$ | $\infty$ | $\infty$ |
| $\tau_2$ | $\sqrt{3}$ | $1$ | $2 + \sqrt{3}$ | $\infty$ | $\infty$ | $\infty$ |
| $\tau_3$ | $1$ | $2 + \sqrt{3}$ | $\sqrt{3}$ | $\infty$ | $\infty$ | $\infty$ |
| $\tau_4$ | $\infty$ | $\infty$ | $\infty$ | $2 + \sqrt{3}$ | $\sqrt{3}$ | $1$ |
| $\tau_5$ | $\infty$ | $\infty$ | $\infty$ | $\sqrt{3}$ | $1$ | $2 + \sqrt{3}$ |
| $\tau_6$ | $\infty$ | $\infty$ | $\infty$ | $1$ | $2 + \sqrt{3}$ | $\sqrt{3}$ |

*Table 4.* An Instance in which all solutions fail $(\frac{1 + \sqrt{3}}{2} - \varepsilon)$-JR

regions, separated by an infinite distance. Within each region, the internal distance structure remains identical in the metric space. We note that the TrSP solutions $\{\tau_1, \tau_2, \tau_3\}$ and $\{\tau_4, \tau_5, \tau_6\}$ fail to provide any approximation of JR, as all three

agents have strong incentives to deviate to any alternative solution that selects at least one transit stop from each region. Such a deviation reduces their cost from infinity to a finite constant.

Due to the identical distance structure in the two regions, without loss of generality, it suffices to consider two TrSP solutions: $Y_1 = \{\tau_1, \tau_2, \tau_6\}$ and $Y_2 = \{\tau_1, \tau_2, \tau_4\}$.[10] Recall that $\lceil \frac{2n}{k} \rceil = \frac{2 \cdot 3}{3} = 2$.

For the solution $Y_1$, consider the deviating coalition $S_1 = \{2, 3\}$ and the alternative set of transit stops $T_1 = \{\tau_1, \tau_4\}$. For agent 2, we have $c_2(Y_1) = d(a_2, \tau_2) + d(b_2, \tau_6) = 1 + 2 + \sqrt{3} = 3 + \sqrt{3}$ and $c_2(T_1) = d(a_2, \tau_1) + d(b_2, \tau_4) = \sqrt{3} + \sqrt{3} = 2\sqrt{3}$. Hence, we obtain $(\frac{1+\sqrt{3}}{2} - \varepsilon) \cdot c_2(T_1) < c_2(Y_1)$. Similarly, for agent 3, we compute $c_3(Y_1) = 1 + \sqrt{3}$ and $c_3(T_1) = 2$, yielding $(\frac{1+\sqrt{3}}{2} - \varepsilon) \cdot c_3(T_1) < c_3(Y_1)$. Thus, $Y_1$ violates $(\frac{1+\sqrt{3}}{2} - \varepsilon)$-JR.

Now consider the solution $Y_2$, with deviating coalition $S_2 = \{1, 2\}$ and alternative transit stops $T_2 = \{\tau_2, \tau_5\}$. For agent 1, we have $c_1(Y_2) = d(a_1, \tau_2) + d(b_1, \tau_4) = \sqrt{3} + 2 + \sqrt{3} = 2 + 2\sqrt{3}$ and $c_1(T_2) = d(a_1, \tau_2) + d(b_1, \tau_5) = 2\sqrt{3}$; For agent 2, we compute $c_2(Y_2) = 1 + \sqrt{3}$ and $c_2(T_2) = 2$. It follows that

$$\min\left\{\frac{c_1(Y_2)}{c_1(T_2)}, \frac{c_2(Y_2)}{c_2(T_2)}\right\} = \min\left\{\frac{2 + 2\sqrt{3}}{2\sqrt{3}}, \frac{1 + \sqrt{3}}{2}\right\} = \frac{1 + \sqrt{3}}{2}.$$

Therefore, $Y_2$ also fails to satisfy $(\frac{1+\sqrt{3}}{2} - \varepsilon)$-JR. $\qquad\square$

**Proposition 4.2.** *For any $\gamma, \rho \geq 1$, there exists a TrSP instance in which ECA fails $(\gamma, \rho)$-core.*

*Proof.* Fix arbitrary $\gamma, \rho \geq 1$ and fix an integer $r \geq 2$. We define $z = \lceil \frac{r}{r-1} \cdot \gamma + 1 \rceil$ and construct a TrSP instance $\mathcal{I} = \langle N, \mathcal{C}, \{\theta_i\}_{i \in N}, k \rangle$ where

$$|N| = \frac{(z^2 - z) \cdot r}{2}, |\mathcal{C}| = z^2, \text{ and } k = z^2 - z.$$

The instance is based on a complete graph $K_z$ with $z$ vertices, where each vertex $i \in [z]$ represents a candidate transit stop $\tau_i$. For every pair of distinct vertices $i, j \in [z]$, the distance between $\tau_i$ and $\tau_j$ is assumed to be infinite. Along each of the edges $(i, j)$ of $K_z$, there are $r$ agents whose endpoints lie on that edge. Additionally, there are two extra candidate stops, denoted by $\tau_{i,j}$ and $\tau_{j,i}$ located on the edge. To illustrate, consider the edge between $\tau_1$ and $\tau_2$, pictured in Figure 6. For the

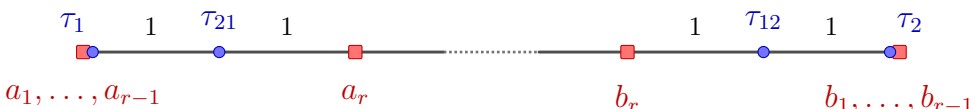

*Figure 6.* One edge in the complete graph $K_z$ with endpoint vertices 1 and 2

edge between $\tau_1$ and $\tau_2$, there are two extra candidate stops $\tau_{12}$ and $\tau_{21}$. All of these 4 candidate stops are marked as blue circles. On this edge, there are $r$ agents with travel endpoints located along the line. Specifically, for agents $1, 2, \ldots, r-1$, the starting point is at $\tau_1$, and terminal point is at $\tau_2$. Agent $r$ has a starting point at $a_r$ and a terminal point at $b_r$. All endpoints are marked as red squares.

The total number of agents is $n = r \cdot \frac{z(z-1)}{2}$, since there are $\frac{z(z-1)}{2}$ edges in $K_z$, with $r$ agents placed on each edge. The total number of candidate transit stops is $z + z(z-1) = z^2$, which includes the $z$ vertices of the complete graph and two additional stops per edge. Given $k = z^2 - z$, we have $\lceil \frac{2n}{k} \rceil = r$. According to the execution procedures of ECA, when the cost radius reaches 2, each cost ball centered around a stop pair $\{\tau_{ij}, \tau_{ji}\}$ will capture all $r$ agents on the corresponding edge $(i, j)$. Therefore, ECA outputs the solution $Y = \bigcup_{i,j \in [z]: i \neq j} \{\tau_{ij}, \tau_{ji}\}$, which contains $2 \cdot \frac{z(z-1)}{2} = z^2 - z = k$ stops and is thus a feasible solution. Now, consider the subset of agents $S$ whose endpoints are on the vertices of $K_z$, that is, excluding agent $r$ on each edge. We have $|S| = \frac{z(z-1)}{2} \cdot (r-1)$. Let $T = \{\tau_1, \tau_2, \ldots, \tau_z\}$ be a deviation with $|T| = z$. For

---

[10] To see this, note that $\tau_4, \tau_5$, and $\tau_6$ mirror $\tau_1, \tau_2$, and $\tau_3$, respectively, and that one of these groups will have exactly one selected stop. Thus, we only need to consider two types of solutions: one in which that selected stop is the counterpart of a selected stop on the other side ($Y_2$), and one in which it is not ($Y_1$).

each agent $i \in S$, since both endpoints coincide with stops in $T$, it holds that $c_i(T) = 0$. On the other hand, under ECA solution $Y$, all agents in $S$ incur cost $c_i(Y) = 2$. Recalling that $z = \lceil \frac{r}{r-1} \cdot \gamma + 1 \rceil$, we see that

$$\frac{|S|}{|T| \cdot \lceil \frac{n}{k} \rceil} = \frac{\frac{z(z-1) \cdot (r-1)}{2}}{z \cdot \frac{r}{2}} = \frac{r-1}{r} \cdot (z-1) \geq \gamma.$$

Therefore, in this instance, under the ECA solution $Y$, there exists a group of agents $S$ and a solution $T$ such that $|S| \geq \gamma \cdot |T| \cdot \frac{n}{k}$, and for every agent $i \in S$, $\rho \cdot c_i(T) = 0 < c_i(Y)$, which implies that ECA fails to satisfy $(\gamma, \rho)$-core for any $\gamma, \rho \geq 1$. $\qquad \square$

**Theorem 4.1.** *For any arbitrary transit cost function $d'(\cdot) \geq 0$, ECA satisfies $(1 + \sqrt{2})$-JR. However, for any $\varepsilon > 0$, there exists an instance with null transit costs for which ECA violates $(1 + \sqrt{2} - \varepsilon)$-JR.*

*Proof.* Given any TrSP instance $\mathcal{I}$, let $Y \subseteq \mathcal{C}$ be the solution computed by ECA. Suppose, for a contradiction, that $Y$ violates $(1 + \sqrt{2})$-JR. It follows that there exists a group of agents $S \subseteq N$ with $|S| \geq \lceil \frac{2n}{k} \rceil$ and a pair of transit stops $T = \{\tau_1, \tau_2\} \subseteq \mathcal{C}$ such that $(1 + \sqrt{2}) \cdot c_j(T) < c_j(Y)$ for all $j \in S$. We first observe that for each agent $j \in S$, it must hold that $c_j(T) < d(a_j, b_j)$, since otherwise $c_j(Y) \leq c_j(T)$. In other words, every agent in $S$ uses the transit stops in $T$ for their route. Without loss of generality, for each agent $j \in S$, denote $a_j$ as the endpoint that uses stop $\tau_1$ and $b_j$ as the endpoint that uses stop $\tau_2$.

Let $r_T = \max_{j \in S} c_j(T)$ be the maximum cost to any agent in $S$ incurred by using transit stops $T$ and let $i^*$ be the agent in $S$ that realizes this maximum. That is, $r_T = d(a_{i^*}, \tau_1) + d'(\tau_1, \tau_2) + d(b_{i^*}, \tau_2)$. If the cost radius considered by ECA never reaches $r_T$, then ECA returns stop placement $Y$ covering all the agents in $N$ with a cost radius smaller than $r_T$, which means that $c_{i^*}(Y) \leq c_{i^*}(T)$, yielding a contradiction.

The other case is that ECA does consider a cost radius of $r_T$ at some point during its execution. In this situation, ECA must add some transit stop pair which gives some agent in $S$ a cost upper bounded by $r_T$ before it continues smoothly increasing the cost radius. To see this, consider that otherwise ECA will add the pair $T = (\tau_1, \tau_2)$ into the solution as $|S| \geq \lceil \frac{2n}{k} \rceil$ and $r_T = \max_{j \in S} c_j(T)$. Therefore, there exists an agent $i \in S$ with $c_i(Y) \leq r_T$. We next prove an upper bound on the cost of $i^*$ under solution $Y$. In particular, we show $c_{i^*}(Y) \leq 2 \cdot r_T + c_i(T)$ by case analysis on agent $i$'s cost under $Y$.

**Case (a).** $c_i(Y) = d(a_i, b_i)$, i.e., agent $i$ walks under $Y$. We prove the statement as follows:

$$
\begin{aligned}
c_{i^*}(Y) &\leq d(a_{i^*}, b_{i^*}) \\
&\leq d(a_{i^*}, \tau_1) + d(\tau_1, a_i) + d(a_i, b_i) + d(b_i, \tau_2) + d(\tau_2, b_{i^*}) && \text{(Triangle inequality)} \\
&\leq \big(d(a_{i^*}, \tau_1) + d'(\tau_1, \tau_2) + d(\tau_2, b_{i^*})\big) + \big(d(\tau_1, a_i) + d'(\tau_1, \tau_2) + d(b_i, \tau_2)\big) + d(a_i, b_i) && (d'(\tau_1, \tau_2) \geq 0) \\
&= c_{i^*}(T) + c_i(T) + d(a_i, b_i) && (c_i(T) = d(a_i, \tau_1) + d'(\tau_1, \tau_2) + d(\tau_2, b_i), \forall i \in S) \\
&= c_{i^*}(T) + c_i(T) + c_i(Y) && (c_i(Y) = d(a_i, b_i)) \\
&\leq 2 \cdot r_T + c_i(T). && (c_{i^*}(T) = r_T, c_i(Y) \leq r_T)
\end{aligned}
$$

**Case (b).** $c_i(Y) < d(a_i, b_i)$, i.e., agent $i$ uses the transit system under $Y$. Let $(y_1, y_2)$ denote the pair of transit stops that agent $i$ uses for minimizing her traveling cost, i.e., $(y_1, y_2) = \arg\min_{y, y' \in Y} d(a_i, y) + d'(y, y') + d(b_i, y')$. The upper bound follows from a similar argument to the previous case.

$$
\begin{aligned}
c_{i^*}(Y) &\leq d(a_{i^*}, y_1) + d'(y_1, y_2) + d(b_{i^*}, y_2) \\
&\leq \big(d(a_{i^*}, \tau_1) + d(\tau_1, a_i) + d(a_i, y_1)\big) + \big(d(b_{i^*}, \tau_2) + d(\tau_2, b_i) + d(b_i, y_2)\big) + d'(y_1, y_2) && \text{(Triangle inequality)} \\
&\leq c_{i^*}(T) + c_i(T) + (d(a_i, y_1) + d(b_i, y_2)) + d'(y_1, y_2) && (c_i(T) \leq d(a_i, \tau_1) + d(\tau_2, b_i), \forall i \in S) \\
&= c_{i^*}(T) + c_i(T) + c_i(Y) && (c_i(Y) = d(a_i, y_1) + d'(y_1, y_2) + d(b_i, y_2)) \\
&\leq 2 \cdot r_T + c_i(T). && (c_{i^*}(T) = r_T, c_i(Y) \leq r_T)
\end{aligned}
$$

Lastly, with the upper bound of $c_{i^*}(Y)$ in hand, we consider the minimum multiplicative cost improvement of agents $i$ and

$i^*$ under $T$:

$$\min\left(\frac{c_i(Y)}{c_i(T)}, \frac{c_{i^*}(Y)}{c_{i^*}(T)}\right) \leq \min\left(\frac{r_T}{c_i(T)}, \frac{2 \cdot r_T + c_i(T)}{r_T}\right) \leq \max_{z \geq 1}(\min(z, 2 + 1/z)) \leq 1 + \sqrt{2},$$

which contradicts that $(1 + \sqrt{2}) \cdot c_j(T) < c_j(Y)$ for all $j \in S$.

To show the tightness, we provide an instance in which ECA fails to achieve $(1 + \sqrt{2} - \varepsilon)$-JR. Fix $\epsilon > 0$ and consider a TrSP instance with 4 agents, 4 candidate stops, and $k = 3$. The transit cost function satisfies $d'(i, j) = 0$ for any $i, j \in \mathcal{C}$. For each $i \in \{1, 2, 3, 4\}$, agent $i$ travels from $a_i$ to $b_i$. The distances between candidate stops and endpoints are specified in Table 5.

| $d(\cdot)$ | $a_1$ | $a_2, a_3$ | $a_4$ | $b_1$ | $b_2, b_3$ | $b_4$ |
|---|---|---|---|---|---|---|
| $\tau_1$ | $1$ | $\sqrt{2} - 1$ | $1 + \sqrt{2}$ | $\infty$ | $\infty$ | $\infty$ |
| $\tau_2$ | $1 + \sqrt{2}$ | $1 - \frac{\varepsilon}{4}$ | $1 - \frac{\varepsilon}{4}$ | $\infty$ | $\infty$ | $\infty$ |
| $\tau_3$ | $\infty$ | $\infty$ | $\infty$ | $1$ | $\sqrt{2} - 1$ | $1 + \sqrt{2}$ |
| $\tau_4$ | $\infty$ | $\infty$ | $\infty$ | $1 + \sqrt{2}$ | $1 - \frac{\varepsilon}{4}$ | $1 - \frac{\varepsilon}{4}$ |

*Table 5.* A TrSP instance in which ECA fails $(1 + \sqrt{2} - \varepsilon)$-JR

We first observe that $\{\tau_2, \tau_4\}$ is the first pair selected by ECA as when the cost radius reaches $2(1 - \frac{\varepsilon}{4})$, agents $\{2, 3, 4\}$ are deactivated by $\{\tau_2, \tau_4\}$. Afterward, no other candidate stop can be selected by ECA. Therefore, ECA returns $\{\tau_2, \tau_4\}$ as the output. However, consider a deviation coalition $S = \{1, 2, 3\}$ and an alternative pair $T = \{\tau_1, \tau_3\}$. For agent 1, we have $c_1(Y) = 2(1 + \sqrt{2})$ and $c_1(T) = 2$. Thus, we have $\frac{c_1(Y)}{c_1(T)} = 1 + \sqrt{2} > 1 + \sqrt{2} - \varepsilon$. For agents 2 and 3, we have $c_2(Y) = c_3(Y) = 2 - \frac{\varepsilon}{2}$ and $c_2(T) = c_3(T) = 2(\sqrt{2} - 1)$. Thus, we have

$$\frac{c_2(Y)}{c_2(T)} = \frac{c_3(Y)}{c_3(T)} = \frac{2 - \frac{\varepsilon}{2}}{2(\sqrt{2} - 1)} = (1 + \sqrt{2}) - \frac{(\sqrt{2} + 1)\varepsilon}{4} > (1 + \sqrt{2}) - \varepsilon.$$

Thus, ECA fails to satisfy $(1 + \sqrt{2} - \varepsilon)$-JR for any $\varepsilon > 0$. $\qquad\square$

## E. $\lambda$-Hybrid: Balancing Core and JR Approximations

As we saw in Sections 2.2 and 4, ECA guarantees the best-known approximation to JR despite performing arbitrarily poorly with respect to core, while GC-TrSP obtains a worse approximation to JR but guarantees a constant-factor approximation to core. In this section, we present an algorithm, parameterized by $\lambda \in [0, 1]$, which effectively navigates the tradeoff between JR and core delineated by ECA and GC-TrSP.[11] Intuitively, this algorithm integrates the decision-making principles of both GC-TrSP and ECA by concurrently simulating both algorithms and considering both individual transit stop candidates and pairs of stops. We call it the $\lambda$-Hybrid algorithm and give a formal description in Algorithm 3. In essence, the parameter $\lambda$ allows tuning between ECA and GC-TrSP by controlling the rates of growth of the respective "radii" of each algorithm relative to each other. Specifically, $\lambda$ encodes the ratio between the rate of growth of the distance radius (GC-TrSP radius) and the cost radius (ECA radius). If the rates of growth are close to equal ($\lambda$ close to 1), the algorithm is closer to GC-TrSP since as the distance radius will likely dominate stop selection. On the other hand, as $\lambda$ approaches 0, the distance radius grows much slower than the cost radius, and the algorithm moreso mimics the behavior of ECA.

We remark that the 0-hybrid algorithm is not equivalent to ECA. To see this, when $\lambda = 0$, note that when given instances where $2n/k$ endpoints are located at the same position, the 0-hybrid algorithm will deactivate these endpoints immediately, as they are already within distance radius zero. In contrast, this location is not guaranteed to be selected by ECA when no pair of stops with 0 cost exists.

In the remainder of this section, we will show that the $\lambda$-Hybrid algorithm offers a way of smoothly navigating the tradeoff

---

[11]The tradeoff we remark on here is purely between these two algorithms. Theoretical evidence of such a tradeoff, for example showing the impossibility of algorithms which guarantee $\alpha$-JR and $(\beta, \gamma)$-core for some $\alpha, \beta, \gamma$, is an interesting direction for future research.

---

**Algorithm 3** $\lambda$-Hybrid

---

**Input:** TrSP instance $\mathcal{I} = \langle N, \mathcal{C}, \{\theta_i\}_{i \in N}, k \rangle$, $\lambda \in [0, 1]$.
**Output:** $Y$.
1: Initialize $r \leftarrow 0$, $Y \leftarrow \emptyset$.
2: Let $\Theta$ be a multiset including all the endpoints of agents in $N$.
3: **while** $\Theta \neq \emptyset$ **do**
4:     Smoothly increase $r$.
5:     **while** $\exists \{a_i, b_i\} \subseteq \Theta$ such that $c_i(Y) \leq r$ or $\exists e_j \in \Theta$ such that $d(e_j, Y) \leq \lambda \cdot r$ **do**
6:         $\Theta \leftarrow \Theta \setminus \{a_i, b_i\}$ or $\Theta \leftarrow \Theta \setminus \{e_j\}$
7:     **end while**
8:     **while** $\exists (\tau_1, \tau_2) \in \mathcal{C}^2 \setminus Y^2$ and $\exists S \subseteq \mathcal{N}, |S| \geq \lceil \frac{2 \cdot n}{k} \rceil$, such that $\forall j \in S, (1) \{a_i, b_i\} \subseteq \Theta$; and $(2) c_j(Y \cup \{\tau_1, \tau_2\}) \leq$
        $r$ **do**
9:         $Y \leftarrow Y \cup \{\tau_1, \tau_2\}$
10:        $\Theta \leftarrow \Theta \setminus \{a_i, b_i\}_{i \in S}$
11:    **end while**
12:    **while** $\exists \tau_3 \in \mathcal{C} \setminus Y$ and $\exists E \subseteq \Theta, |E| \geq \lceil \frac{2n}{k} \rceil$, such that $\forall e \in E, \ d(e, Y \cup \{\tau_3\}) \leq \lambda \cdot r$ **do**
13:        $Y \leftarrow Y \cup \{\tau_3\}$
14:        $\Theta \leftarrow \Theta \setminus E$
15:    **end while**
16: **end while**
17: Return $Y$.

---

between JR and core created by ECA and GC-TrSP. Specifically, we show that $\lambda$-Hybrid satisfies

$$\frac{\lambda + 3 + \sqrt{\lambda^2 + 10\lambda + 9}}{2}\text{-JR and } \left(2, \frac{\sqrt{\lambda^2 + 6\lambda + 1} + \lambda + 1}{2\lambda}\right)\text{-core}$$

where the JR approximation upper bound holds for $\lambda \in [0, 1]$ and the core approximation ratio holds for all $\lambda \in (0, 1]$. Figure 7 plots the fairness approximations obtained as a function of $\lambda$, showing that, as $\lambda$ increases, the approximation ratio of JR worsens while that of core improves, as expected.

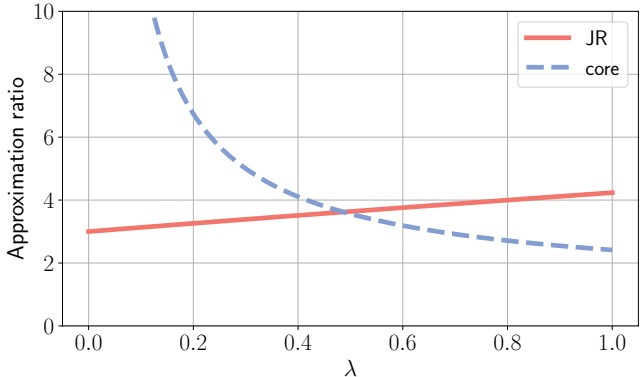

*Figure 7.* Parameter $\lambda \in [0, 1]$. Red solid line represents the JR approximation ratio of $\frac{\lambda + 3 + \sqrt{\lambda^2 + 10\lambda + 9}}{2}$ and blue dashed line represents the parameterized function $\frac{\sqrt{\lambda^2 + 6\lambda + 1} + \lambda + 1}{2\lambda}$ of core approximation.

## E.1. JR analysis of $\lambda$-Hybrid

Parameterized by $\lambda \in [0, 1]$, we begin by analyzing the extent to which the $\lambda$-Hybrid algorithm approximates JR. Building on the ideas underlying the JR analysis of the GC-TrSP algorithm and ECA, we establish that the $\lambda$-Hybrid algorithm achieves a $\frac{\lambda + 3 + \sqrt{\lambda^2 + 10\lambda + 9}}{2}$-approximation of JR. As $\lambda$ approaches 0, the $\lambda$-Hybrid algorithm aligns more closely with ECA, thereby attaining a stronger approximation of JR. Conversely, as $\lambda$ approaches 1, the algorithm shifts towards the

behavior of GC-TrSP, which yields a weaker JR approximation but, in turn, provides a stronger guarantee with respect to core approximation (see the next subsection). For ease of exposition, for the remainder of the section, we will refer to the selection process in lines 8-11 as the "ECA loop" and the selection process in lines 12-15 as the "GC-TrSP loop".

**Theorem E.1.** $\lambda$*-Hybrid satisfies* $\frac{\lambda+3+\sqrt{\lambda^2+10\lambda+9}}{2}$*-JR, where* $\lambda \in [0, 1]$*, and this bound is tight.*

*Proof.* Let $Y \subseteq \mathcal{C}$ be the transit stop solution returned by $\lambda$-Hybrid. Note that $\frac{\lambda+3+\sqrt{\lambda^2+10\lambda+9}}{2} \geq 3$ on the interval $[0, 1]$. Suppose for a contradiction that $Y$ violates $\frac{\lambda+3+\sqrt{\lambda^2+10\lambda+9}}{2}$-JR. That is, there exists a group of agents $S \subseteq N, |S| \geq \lceil \frac{2n}{k} \rceil$, and pair of transit stops $T \subseteq \mathcal{C}$ such that for every agent $i \in S$, $\frac{\lambda+3+\sqrt{\lambda^2+10\lambda+9}}{2} \cdot c_i(T) < c_i(Y)$. Denote the two stops in $T$ by $\tau_1$ and $\tau_2$. For each agent $i \in S$, there is a matching between their endpoints and stops $\tau_1, \tau_2$. Without loss of generality, for each agent $i \in S$, denote $a_i$ as the endpoint that uses stop $\tau_1$ and $b_i$ as the endpoint that uses stop $\tau_2$. Let $r_T = \max_{i \in S} c_i(T)$ be the maximum cost of any agent in $S$ when using transit stop pair $T = \{\tau_1, \tau_2\}$. Let $i^*$ be an agent in $S$ that realizes this maximum distance, i.e., $r_T = c_{i^*}(T)$.

We first consider the case when the parameter $r$ never reaches $r_T$. That is, all endpoints are deactivated either by the GC-TrSP loop with a distance radius at most $\lambda \cdot r_T$ or by the ECA loop with a cost radius at most $r_T$. Consequently, we have $c_{i^*}(Y) \leq \max(2\lambda \cdot r_T, r_T) \leq 2 \cdot c_{i^*}(T)$, contradicting that $\frac{\lambda+3+\sqrt{\lambda^2+10\lambda+9}}{2} \cdot c_{i^*}(T) < c_{i^*}(Y)$.

We next consider the case in which the parameter $r$ reaches $r_T$ (i.e., the GC-TrSP loop reaches distance radius $\lambda \cdot r_T$ and the ECA loop reaches cost radius $r_T$). Notice that the algorithm reaches radius parameter $r_T$ but does not select $(\tau_1, \tau_2)$. This implies that either (a) an agent in $S$, or (b) an endpoint of an agent in $S$, has already been deactivated during the execution of the algorithm.

**Case (a).** There exists an agent $j \in S$ such that $j$ is deactivated by some pair of transit stops $\{y_1, y_2\} \subseteq Y$ in the ECA loop with a cost radius of at most $r_T$. Therefore, it holds that $c_j(Y) \leq r_T$. For agent $i^*$, we have

$$
\begin{aligned}
c_{i^*}(Y) &\leq d(a_{i^*}, y_1) + d(b_{i^*}, y_2) \\
&\leq d(a_{i^*}, \tau_1) + d(\tau_1, a_j) + d(a_j, y_1) + d(b_{i^*}, \tau_2) + d(\tau_2, b_j) + d(b_j, y_2) \\
&\leq 2 \cdot r_T + c_j(T).
\end{aligned}
$$

The minimum multiplicative cost improvement for agents $i^*$ and $j$ is:

$$
\min\left(\frac{c_j(Y)}{c_j(T)}, \frac{c_{i^*}(Y)}{c_{i^*}(T)}\right) \leq \min\left(\frac{r_T}{c_j(T)}, \frac{2 \cdot r_T + c_j(T)}{r_T}\right) \leq \max_{z \geq 0}\left(\min\left(z, 2 + \frac{1}{z}\right)\right) \leq 1 + \sqrt{2},
$$

again contradicting that $\frac{\lambda+3+\sqrt{\lambda^2+10\lambda+9}}{2} \cdot c_i(T) < c_i(Y)$ for all $i \in S$.

**Case (b).** There exists an endpoint of an agent $j \in S$ which is deactivated by some singleton transit stop candidate $y_1$ in the GC-TrSP loop with a distance radius at most $\lambda \cdot r_T$. The tougher subcase is that in which $\tau_2 \notin Y$. Notice that $\tau_2$ can deactivate $|S| \geq \frac{2n}{k}$ terminal endpoints for agents in $S$ with radius at most $r_T$, but is not selected by the algorithm. Hence, there must exist some selected transit stop $y_2 \in Y$ which deactivates at least one terminal endpoint of some agent in $S$ with radius no greater than $r_T$. Denote the corresponding agent and endpoint by $j'$ and $b_{j'}$. We first consider the cost of agent $j$ under $Y$,

$$
\begin{aligned}
c_j(Y) &\leq d(a_j, y_1) + d(b_j, y_2) \\
&\leq \lambda \cdot r_T + d(b_j, \tau_2) + d(\tau_2, b_{j'}) + d(b_{j'}, y_2) \\
&\leq (\lambda + 1) \cdot r_T + d(b_j, \tau_2) + d(\tau_2, b_{j'}).
\end{aligned}
$$

Similarly, we consider the cost of agent $j'$ under $Y$,

$$
\begin{aligned}
c_{j'}(Y) &\leq d(a_{j'}, y_1) + d(b_{j'}, y_2) \\
&\leq d(a_{j'}, \tau_1) + d(\tau_1, a_j) + d(a_j, y_1) + d(b_{j'}, y_2) \\
&\leq (\lambda + 1) \cdot r_T + d(\tau_1, a_j) + d(a_{j'}, \tau_1).
\end{aligned}
$$

We next focus on agent $i^*$ and show the following upper bound:

$$c_{i^*}(Y) \leq d(a_{i^*}, y_1) + d(b_{i^*}, y_2)$$
$$\leq d(a_{i^*}, \tau_1) + d(\tau_1, a_j) + d(a_j, y_1) + d(b_{i^*}, \tau_2) + d(\tau_2, b_{j'}) + d(b_{j'}, y_2)$$
$$\leq r_T + d(\tau_1, a_j) + \lambda \cdot r_T + d(\tau_2, b_{j'}) + r_T$$
$$\leq (\lambda + 2) \cdot r_T + d(\tau_1, a_j) + d(\tau_2, b_{j'}).$$

To clarify the expression, let $d(\tau_1, a_j)$ be $x$ and $d(b_{j'}, \tau_2)$ be $y$. With these upper bounds in hand, we derive the minimum multiplicative cost improvement of agents $\{i^*, j, j'\}$ under $T$.

$$\min\left(\frac{c_j(Y)}{c_j(T)}, \frac{c_{j'}(Y)}{c_{j'}(T)}, \frac{c_{i^*}(Y)}{c_{i^*}(T)}\right) \leq \min\left(\frac{(\lambda+1)\cdot r_T + d(b_j, \tau_2) + y}{x + d(b_j, \tau_2)}, \frac{(\lambda+1)\cdot r_T + x + d(a_{j'}, \tau_1)}{d(a_{j'}, \tau_1) + y}, \frac{(\lambda+2)\cdot r_T + x + y}{r_T}\right)$$
$$\leq \min\left(\frac{(\lambda+1)\cdot r_T + y}{x}, \frac{(\lambda+1)\cdot r_T + x}{y}, \frac{(\lambda+2)\cdot r_T + x + y}{r_T}\right).$$

To upper-bound this expression, we consider

$$\min\left(\frac{(\lambda+1)\cdot r_T + y}{x}, \frac{(\lambda+1)\cdot r_T + x}{y}, (\lambda+2) + \frac{x+y}{r_T}\right) \leq \min\left(1 + \frac{(\lambda+1)\cdot r_T}{x}, (\lambda+2) + \frac{2x}{r_T}\right)$$
$$\leq \max\left(\min_{q\geq 1}\left(1 + (\lambda+1)q, (\lambda+2) + \frac{2}{q}\right)\right)$$
$$= \frac{\lambda + 3 + \sqrt{\lambda^2 + 10\lambda + 9}}{2},$$

where it holds with equality when $q = \frac{\lambda + 1 + \sqrt{\lambda^2 + 10\lambda + 9}}{2(\lambda+1)}$. Therefore, we conclude that for some agent in $i' \in \{j, j', i^*\} \subseteq S$, we have $c_{i'}(Y) \leq \frac{\lambda + 3 + \sqrt{\lambda^2 + 10\lambda + 9}}{2} \cdot c_{i'}(T)$, contradicting to the assumption that for every agent $i \in S$, $\frac{\lambda + 3 + \sqrt{\lambda^2 + 10\lambda + 9}}{2} \cdot c_i(T) < c_i(Y)$.

Lastly, we just need to handle the easier subcase in which $\tau_2$ already belongs to the stop placement, i.e. $\tau_2 \in Y$. Here we need only consider the multiplicative improvement of agents $j$ and $i^*$:

$$\min\left(\frac{c_j(Y)}{c_j(T)}, \frac{c_{i^*}(Y)}{c_{i^*}(T)}\right) \leq \min\left(\frac{\lambda \cdot r_T + d(b_j, Y)}{d(a_j, \tau_1) + d(b_j, \tau_2)}, \frac{d(a_{i^*}, \tau_1) + d(a_j, \tau_1) + \lambda \cdot r_T + d(b_{i^*}, Y)}{c_{i^*}(T)}\right)$$
$$\leq \min\left(\frac{\lambda \cdot r_T + d(b_j, \tau_2)}{d(a_j, \tau_1) + d(b_j, \tau_2)}, \frac{c_{i^*}(T) + d(a_j, \tau_1) + \lambda \cdot r_T}{r_T}\right)$$
$$\leq \min\left(\frac{\lambda \cdot r_T}{d(a_j, \tau_1)}, \frac{(\lambda+1)r_T + d(a_j, \tau_1)}{r_T}\right)$$
$$\leq \max\left(\min_{q\geq 1}\left(\lambda \cdot q, 1 + \lambda + 1/q\right)\right)$$
$$= \frac{\lambda + 1 + \sqrt{\lambda^2 + 6\lambda + 1}}{2}.$$

Since this value never exceeds $1 + \sqrt{2}$ on the unit interval, this also provides a contradiction and concludes our upper bound proof.

To show the tightness of this approximation ratio, we slightly modify the instance originally used to prove the tightness of the GC-TrSP algorithm. In the example, we have $n = 7$ and $k = 4$ with candidate stops $\mathcal{C} = \{\tau_1, \tau_2, y_1, y_2\} \cup D$ where $D = \{y_3, y_4, y_5, y_6\}$. For candidate stops in $D$, we assign a large constant distance to each endpoint. The locations of endpoints and their distances (except candidate stops in $D$) are illustrated in Figure 8.

Consider the execution of the $\lambda$-Hybrid algorithm, keeping in mind that $\lceil 2n/k \rceil = 4$. Since $\lambda \in [0, 1]$, we first observe that $y_1$ is the first selected transit stop as when the parameter $r$ reaches $1 - \frac{\varepsilon}{2}$, $y_1$ will deactivate endpoints $\{a_4, a_5, a_6, a_7\}$ with a distance radius of $\lambda \cdot (1 - \frac{\varepsilon}{2})$ via the GC-TrSP loop. Notice that the pair $\{\tau_1, \tau_2\}$ are not selected because they can deactivate 4 agents only when the parameter $r$ reaches 1. After the selection of $y_1$, it is not until the parameter $r$ reaches

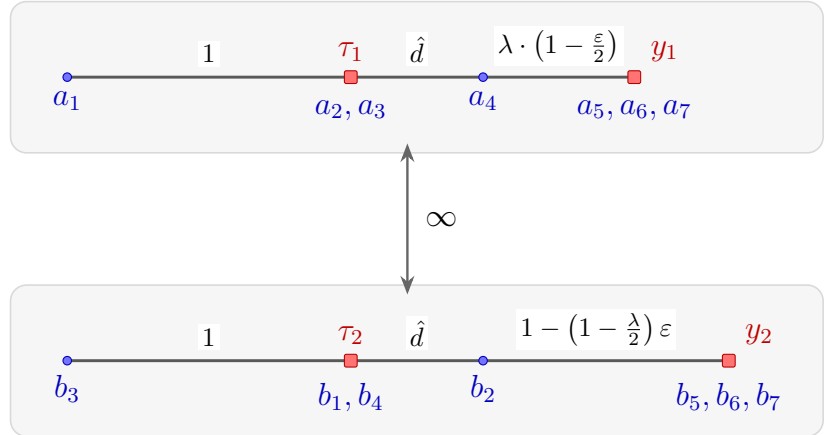

*Figure 8.* TrSP instance where $\lambda$-Hybrid violates $(\frac{\lambda+3+\sqrt{\lambda^2+10\lambda+9}}{2} - \varepsilon)$-JR where the distance $\hat{d} = \frac{\sqrt{\lambda^2+10\lambda+9}-\lambda-1}{4}$, which is in the range of $[\frac{1}{2}, \frac{\sqrt{5}-1}{2}]$ for $\lambda \in [0, 1]$.

$(1 - (1 - \frac{\varepsilon}{2}) \cdot \varepsilon)/\lambda < \frac{1}{\lambda}$ that candidate stop $y_2$ deactivates 4 endpoints $\{b_2, b_5, b_6, b_7\}$ in a "GC-TrSP" loop. Notice that after the selection of $y_1$ and $y_2$, no other candidate stop or pair is selected by $\lambda$-Hybrid algorithm as all the endpoints will be deactivated when the parameter $r$ reaches $\lambda \cdot (\hat{d} + 2 - (1 - \frac{\lambda}{2}) \cdot \varepsilon)$. Hence, $\lambda$-Hybrid algorithm finally outputs $Y = \{y_1, y_2\}$.

Let $T$ denote the pair $\{\tau_1, \tau_2\}$ and consider agent subset $S = \{1, 2, 3, 4\}$. For agent 1, the cost of using $Y$ is $c_1(Y) = \frac{\lambda+3+\sqrt{\lambda^2+10\lambda+9}}{2} - \varepsilon$ while the cost of using $T$ is $c_1(T) = 1$. Then we have $\frac{c_1(Y)}{c_1(T)} = \frac{\lambda+3+\sqrt{\lambda^2+10\lambda+9}}{2} - \varepsilon$. Similarly, we have $\frac{c_3(Y)}{c_3(T)} = \frac{\lambda+3+\sqrt{\lambda^2+10\lambda+9}}{2} - \varepsilon$ as agent 3 shares the same cost of using $Y$ and $T$ as agent 1.

For agent 2 (the same for agent 4), we compute that $c_2(Y) = \hat{d} + \lambda \cdot (1 - \frac{\varepsilon}{2}) + 1 - (1 - \frac{\lambda}{2}) \cdot \varepsilon = 1 + \hat{d} + \lambda - \varepsilon$ and $c_2(T) = \hat{d}$. Consequently, we have

$$
\begin{aligned}
\frac{c_2(Y)}{c_2(T)} &= \frac{1 + \hat{d} + \lambda - \varepsilon}{\hat{d}} = 1 + \frac{1 + \lambda}{\hat{d}} - \frac{\varepsilon}{\hat{d}} \\
&= 1 + \frac{4(1 + \lambda)}{\sqrt{\lambda^2 + 10\lambda + 9} - \lambda - 1} - \frac{\varepsilon}{\hat{d}} \\
&= 1 + \frac{4(1 + \lambda) \cdot (\sqrt{\lambda^2 + 10\lambda + 9} + (\lambda + 1))}{\lambda^2 + 10\lambda + 9 - (\lambda + 1)^2} - \frac{\varepsilon}{\hat{d}} \\
&= 1 + \frac{4(1 + \lambda) \cdot (\sqrt{\lambda^2 + 10\lambda + 9} + (\lambda + 1))}{8(\lambda + 1)} - \frac{\varepsilon}{\hat{d}} \\
&\leq 1 + \frac{\sqrt{\lambda^2 + 10\lambda + 9} + (\lambda + 1)}{2} - \varepsilon \\
&= \frac{\sqrt{\lambda^2 + 10\lambda + 9} + \lambda + 3}{2} - \varepsilon.
\end{aligned}
$$

Hence, we now have a group of agents $S$ with size $\lceil \frac{2n}{k} \rceil$ such that for each agent $i \in S$, it holds that $\frac{c_i(Y)}{c_i(T)} \leq \frac{\sqrt{\lambda^2+10\lambda+9}+\lambda+3}{2} - \varepsilon$. This implies that $\lambda$-Hybrid algorithm fails to satisfy $(\frac{\lambda+3+\sqrt{\lambda^2+10\lambda+9}}{2} - \varepsilon)$-JR. $\square$

### E.2. Analysis of Core Fairness of the $\lambda$-Hybrid Algorithm

We next analyze the core approximation ratio of the $\lambda$-Hybrid algorithm. Our analysis leverages Theorem 3.1, which provides the crucial link between proportional fairness in clustering and core in TrSP. We will demonstrate that the $\lambda$-Hybrid algorithm satisfies $(\frac{\sqrt{\lambda^2+6\lambda+1}+\lambda+1}{2\lambda})$-PF in the induced clustering instance, which immediately implies that it satisfies $(2, \frac{\sqrt{\lambda^2+6\lambda+1}+\lambda+1}{2\lambda})$-core.

**Lemma E.2.** *For any $\lambda \in (0, 1]$ and any TrSP instance $\mathcal{I}$, the $\lambda$-Hybrid solution under $\mathcal{I}$ guarantees $\frac{\sqrt{\lambda^2+6\lambda+1}+\lambda+1}{2\lambda}$-PF in*

*the induced clustering instance $\mathcal{I}^C$.*

*Proof.* Given any TrSP instance $\mathcal{I} = \langle N, \mathcal{C}, k, \{\theta_i\}_{i \in N} \rangle$, let $Y$ be the TrSP solution returned by $\lambda$-Hybrid under $\mathcal{I}$. We show that $Y$ satisfies $\frac{\sqrt{\lambda^2+6\lambda+1}+\lambda+1}{2\lambda}$-PF for the induced clustering instance $\mathcal{I}^C = \langle \Theta, \mathcal{C}, k \rangle$. Suppose, for the sake of contradiction, that there exists a set of datapoints $\theta' \subseteq \Theta$ and candidate center $c \in \mathcal{C}$ such that $|\theta'| \geq 2 \cdot \lceil \frac{n}{k} \rceil$ and $\frac{\sqrt{\lambda^2+6\lambda+1}+\lambda+1}{2\lambda} \cdot d(e, c) < d(e, Y)$ for each $e \in \theta'$.

Let $r_T = \max_{e \in \theta'} d(e, c)$, let $e^*$ denote the point in $\theta'$ that attains this maximum, and let $i^*$ denote the agent which $e^*$ is an endpoint of in the original TrSP instance, i.e., $e^* \in \theta_{i^*}$. We first consider the case in which the parameter $r$ never reaches $r_T/\lambda$ during the execution of $\lambda$-Hybrid. If $e^*$ was deactivated during the GC-TrSP loop, then $d(e^*, Y) \leq \lambda(r_T/\lambda) = r_T = d(e^*, c)$, providing a contradiction. The other sub-case is that $e^*$ was instead deactivated during the ECA loop, in which case it follows that $d(e^*, Y) \leq c_{i^*}(Y) \leq r_T/\lambda = (1/\lambda) \cdot d(e^*, c)$. It can be verified that $\frac{1}{\lambda} \leq \frac{\sqrt{\lambda^2+6\lambda+1}+\lambda+1}{2\lambda}$ for all $\lambda \in (0, 1]$, meaning this also contradicts our assumption.

We now restrict our attention to the case in which the parameter $r$ reaches $r_T/\lambda$ during the execution of $\lambda$-Hybrid. Note that when $r = r_T/\lambda$, if all endpoints in $\theta'$ remain active, the ball of radius $\lambda \cdot (r_T/\lambda) = r_T$ centered at $c$ captures at least $2 \cdot \lceil n/k \rceil$ endpoints. Since $c$ is not selected, there must be at least one datapoint which was deactivated when the parameter $r$ was at most $r_T/\lambda$. We term this endpoint $e'$, denote the agent it belongs to by $i'$, and denote the candidate center selected in the round $e'$ is deactivated by $y$.

We claim that $d(e', y) \leq r_T/\lambda$. To see this, note that if $y$ was selected in a GC-TrSP round, it holds that $d(e', y) \leq \lambda \cdot r_T/\lambda \leq r_T/\lambda$. Otherwise, $y$ was selected in an ECA round, and it follows that $d(e', y) \leq c_{i'}(Y) \leq r_T/\lambda$. Using this, we now obtain a contradiction by considering the minimum multiplicative improvement attained by endpoints $e^*$ and $e'$ from $c$:

$$\min\left(\frac{d(e^*, Y)}{d(e^*, c)}, \frac{d(e', Y)}{d(e', c)}\right) \leq \min\left(\frac{d(e^*, c) + d(c, e') + d(e', y)}{r_T}, \frac{r_T/\lambda}{d(e', c)}\right)$$
$$\leq \min\left(\frac{(1 + 1/\lambda)r_T + d(e', c)}{r_T}, \frac{r_T/\lambda}{d(e', c)}\right)$$
$$\leq \max_{q > 0}\left(\min(q + 1 + 1/\lambda, 1/(\lambda \cdot q))\right)$$
$$= \frac{\sqrt{\lambda^2 + 6\lambda + 1} + \lambda + 1}{2\lambda}.$$

where the final equality holds because the maximum in the penultimate expression is obtained when $q = (\sqrt{\lambda^2 + 6\lambda + 1} - \lambda - 1)/(2\lambda)$. $\square$

Combining Theorem 3.1 and Lemma E.2 yields the following corollary, which gives an upper bound on the core approximation guaranteed by the $\lambda$-Hybrid rules.

**Corollary E.3.** *For every $\lambda \in (0, 1]$, the $\lambda$-hybrid algorithm satisfies $(2, \frac{\sqrt{\lambda^2+6\lambda+1}+\lambda+1}{2\lambda})$-core.*

We give an almost tight lower bound which, when taken together with Remark E.5, gives a bound between that of ECA and that of GC-TrSP, as one would expect.

**Proposition E.4.** *Given any $\lambda \in (0, 1]$ and $\delta, \varepsilon > 0$, there is an instance for which $\lambda$-hybrid does not satisfy $(2 - \delta, \frac{\sqrt{4\lambda^2+12\lambda+1}+2\lambda+1}{4\lambda} - \varepsilon)$-core.*

*Proof.* Let $H = \lceil 2/\delta \rceil$. We construct an instance with $n = 4 \cdot H$ agents, $m = 12$ candidate transit stops $\mathcal{C} = \{\tau_1, \tau_2, \tau_3, \tau_4, c_1, \ldots, c_8\}$, and $k = 8$. The agent set $N$ is partitioned into six subsets, denoted $N_i$ for $i \in [6]$. The number of agents in each group and their start and end locations are given in Table 6.

The distances from the locations mentioned in Table 6 to the candidate stops $\{\tau_1, \tau_2, \tau_3, \tau_4, c_1, c_2, c_3, c_4\}$ are specified in Table 7. For simplicity, we use $q := (\sqrt{4\lambda^2 + 12\lambda + 1} - 2\lambda - 1)/4\lambda$. Values in parentheses correspond to each other and any location-candidate pairs not specified are assigned an infinite distance. For the remaining stops $\{c_5, c_6, c_7, c_8\}$, we assign a large constant distance to each endpoint. Noting that $q \leq 1$ for all $\lambda \in (0, 1]$, one can check that the resulting metric space satisfies the triangle inequality.

| $i$ | $|N_i|$ | $(a_i, b_i)$ |
|---|---|---|
| 1 | $H-1$ | $(x_1, x_3)$ |
| 2 | $H-1$ | $(x_2, x_4)$ |
| 3 | $H-1$ | $(y_1, y_2)$ |
| 4 | $H-1$ | $(y_3, y_4)$ |
| 5 | 2 | $(z_1, z_2)$ |
| 6 | 2 | $(z_3, z_4)$ |

*Table 6.* The number of agents and endpoint locations of the partition of agents, $N$, in the instance used to prove Proposition E.4.

| $d(\cdot)$ | $\tau_1(\tau_3)$ | $c_1(c_3)$ | $\tau_2(\tau_4)$ | $c_2(c_4)$ |
|---|---|---|---|---|
| $x_1(x_3)$ | 1 | $1 + q + 1/(2 \cdot \lambda) - \varepsilon$ | $\infty$ | $\infty$ |
| $y_1(y_3)$ | $q$ | $1/(2\lambda) - q \cdot \varepsilon$ | $\infty$ | $\infty$ |
| $z_1(z_3)$ | $1 + q$ | $1/(2\lambda) - q \cdot \varepsilon$ | $\infty$ | $\infty$ |
| $x_2(x_4)$ | $\infty$ | $\infty$ | 1 | $1 + q + 1/(2 \cdot \lambda) - \varepsilon$ |
| $y_2(y_4)$ | $\infty$ | $\infty$ | $q$ | $1/(2\lambda) - q \cdot \varepsilon$ |
| $z_2(z_4)$ | $\infty$ | $\infty$ | $1 + q$ | $1/(2\lambda) - q \cdot \varepsilon$ |

*Table 7.* Distances for an instance in which $\lambda$-hybrid fails $(2 - \delta, \frac{\sqrt{4\lambda^2 + 12\lambda + 1} + 2\lambda + 1}{4\lambda} - \varepsilon)$-core.

In words, there are four "zones" where endpoints and candidate stops lie proximal to each other and each of these zones has an identical structure of endpoint distances. Each agent has an endpoint in exactly two zones, and notably, $N_1$ and $N_2$ do not share their respective zone pairs with any other agent group, whereas $N_3$ and $N_4$ share their zone pairs with $N_5$ and $N_6$, respectively. We will now explain the execution of $\lambda$-hybrid and show that it selects $Y = \{c_1, c_2, c_3, c_4\}$. Note that $\lceil 2n/k \rceil = H$. Thus, the endpoints located at any single point in the table above are not enough to trigger the GC-TrSP loop. However, any ball capturing at least two of the points in Table 7 *is* sufficient to trigger the GC-TrSP loop.

We begin with the case in which $\lambda > \frac{1}{2(1+\varepsilon)}$. Observe that, for each $j \in [4]$, there are $H + 1$ endpoints within a radius of $1/(2\lambda) - \varepsilon < 1$ of $c_j$ (specifically endpoints located at $y_j$ and $z_j$, whereas 1 is the minimum radius required to capture at least $H$ endpoints with candidate $\tau_j$ for each $j$. Also note that the pair of stops with the smallest cost radius in this case is $c_1$ and $c_2$ ($c_3$ and $c_4$), which capture agents in $N_3 \cup N_5$ ($N_4 \cup N_6$) with a cost radius of $1/\lambda - 2 \cdot q \cdot \varepsilon < 2$. Thus, regardless of whether it is the GC-TrSP loop or ECA loop which acts first in $\lambda$-hybrid, we can assume that either $c_1$ or $c_1$ and $c_2$ are selected first. In the former case, $c_2$, $c_3$, and $c_4$ will remain the stops with the minimum radius which can capture $H$ active endpoints and will thus be selected next by the GC-TrSP loop. In the latter case, $c_3$ and $c_4$ would be selected afterward by the ECA loop as they can capture $H + 1$ agents with an identical cost radius to $c_1$ and $c_2$. This means that $\lambda$-hybrid will certainly select $Y = \{c_1, c_2, c_3, c_4\}$ first. At this point, only endpoints belonging to agents in $N_1$ and $N_2$ remain active. These agents would be deactivated by the stops selected already, since every remaining candidate requires a very large radius to capture both $N_1$ and $N_2$, and $\lambda$-hybrid would return $Y$.

Next, we handle the case in which $\lambda \leq \frac{1}{2(1+\varepsilon)}$. Here, the $c_j$ stops are not favored by the GC-TrSP loop since $1/(2\lambda) - \varepsilon \geq 1$. Instead, the GC-TrSP loop would first select $\tau_j$ for some $j$ and it would do so when the $r$ parameter increases so that $r \cdot \lambda = 1 \implies r = 1/\lambda$. It can be verified that the pair of stops which capture at least $H$ agents with the minimum cost radius is $c_1$ and $c_2$ (or $c_3$ and $c_4$). This holds precisely because agent groups $N_1$ and $N_2$ are located in distinct zone pairs from $N_3$ and $N_4$ and thus it is impossible for multiple of these groups to benefit from the selection of two stops. For example, the selection of $\tau_1$ and $\tau_2$ can capture agents in $N_3$ with a cost radius of $2q$, but incurs infinite cost for agents in $N_1$, $N_2$, and $N_4$. Note that the agents in $N_3 \cup N_5$ all incur a cost of $2(1/(2\lambda) - \varepsilon)$ by using $c_1, c_2$. Thus, the radius parameter $r$ required to select $c_1$ and $c_2$ is strictly less than $1/\lambda$ and hence the ECA loop is triggered first and $c_1$ and $c_2$ are selected. By the same argument, $c_3$ and $c_4$ are selected by the ECA loop as well. Thus, $\lambda$-hybrid selects $Y = \{c_1, c_2, c_3, c_4\}$ first and the same argument as was used in the first case applies to show that this is the set returned by $\lambda$-hybrid.

Now consider the set of agents $S = N_1 \cup N_2 \cup N_3 \cup N_4$ and transit stop set $T = \{\tau_1, \tau_2, \tau_3, \tau_4\}$. For each agent $i \in N_1 \cup N_2$,

it holds that

$$c_i(Y)/c_i(T) = 1 + q + 1/(2 \cdot \lambda) - \varepsilon = \frac{\sqrt{4\lambda^2 + 12\lambda + 1} + 2\lambda + 1}{4\lambda} - \varepsilon,$$

and for each agent $i \in N_3 \cup N_4$, it holds that

$$
\begin{aligned}
c_i(Y)/c_i(T) &= \frac{1/(2\lambda) - q \cdot \varepsilon}{q} \\
&= \frac{2}{\sqrt{4\lambda^2 + 12\lambda + 1} - 2\lambda - 1} - \varepsilon \\
&= \frac{\sqrt{4\lambda^2 + 12\lambda + 1} + 2\lambda + 1}{4\lambda} - \varepsilon
\end{aligned}
$$

where the final equality follows from multiplying the fraction's numerator and denominator by the conjugate of the denominator. Lastly, we have

$$|S| = 4 \cdot H - 4 = 2 \cdot H(2 - \frac{2}{H}) \geq (2 - \delta) \cdot \frac{|T| \cdot n}{k}.$$

In summary, there exists such a blocking coalition $S$ and a candidate stop subset $T$ with size $|S| \geq (2 - \delta) \cdot \frac{|T| \cdot n}{k}$ such that for every agent $i \in S$, we have $(\frac{\sqrt{4\lambda^2 + 12\lambda + 1} + 2\lambda + 1}{4\lambda} - \varepsilon) \cdot c_i(T) < c_i(Y)$.

$\square$

The lower bound implied by Proposition E.4 for $\lambda \geq \sqrt{2}/2$ is weaker than the lower bound proved for GC-TrSP in Proposition 3.2. Given this, we complement Proposition E.4 by strengthening the core lower bound of $\lambda$-hybrid for the case of $\lambda \geq 1/2$ in the following remark.

*Remark* E.5. Given any $\lambda \geq 1/2$, there is an instance for which $\lambda$-hybrid does not satisfy $(2 - \delta, 1 + \sqrt{2} - \varepsilon)$-core. This follows from the exact same example and argument used to prove Proposition 3.2. In particular, due to the symmetry of the instance given by Table 3, as long as $\lambda \geq 1/2$, the ECA loop will not select any candidates in the execution of $\lambda$-hybrid. This means the algorithm will execute identically to GC-TrSP on this instance.

Given Remark E.5, it is likely that the lower bound in Proposition E.4 is not tight. If it is the case that our upper bound is indeed tight, this suggests that the core approximation of $\lambda$-Hybrid is thanks to the algorithm's approximation of PF (rather than any consideration involving cost). Again, we observe that while taking a clustering approach naively ignores agent-specific cost information, it serves as a very useful tool in the pursuit of core approximation.

## F. Comprehensive Experimental Evaluation

In this section, we present additional details of the experimental setup and further experimental results.

### F.1. Detailed Experimental Setup

**Solution Approximation Verification.** The key step regarding the experimental analysis is to compute the approximation ratio of JR and core with respect to the solutions outputted by GC-TrSP and ECA. We generally follow the verification procedures from Bullinger et al. (2025) (Appendix C).

To test whether a given solution satisfies JR and core, we rely on the following idea. Fix a solution and examine whether there exist deviations to pairs of stops that would strictly reduce agents' costs. Such deviations, if sufficiently widespread, witness a violation of JR or of core stability.

Formally, for any solution $Y \subseteq \mathcal{C}$ and agent $i \in N$, define

$$\mathcal{P}_i(Y) := \{T \subseteq \mathcal{C} : |T| = 2, c_i(T) < c_i(Y)\},$$

that is, the set of stop pairs to which agent $i$ can deviate and obtain a strictly lower cost. Using this notation, we recall the Proposition C.1 of Bullinger et al. (2025).

**Proposition F.1.** *Consider a solution $Y \subseteq C$. We have,*

1. *$Y$ satisfies JR if and only if there is no set $T \subseteq C$ with $|T| = 2$ such that $|\{i \in N : T \in \mathcal{P}_i(Y)\}| \geq \frac{2n}{k}$.*

2. *$Y$ is in the core if and only if there is no set $T \subseteq C$ with $T \neq \emptyset$ such that $|i \in N : \exists\, T' \in \mathcal{P}_i(Y), T' \subseteq T| \geq \frac{|T| \cdot n}{k}$.*

Given a solution $Y$, checking whether it satisfies JR can be done in polynomial time. Specifically, one can compute $\mathcal{P}_i(Y)$ for each agent $i \in N$, and then verify the condition in Proposition F.1 (1) by scanning over stop pairs $T$ and counting how many agents include $T$ in their deviation pairs. Moreover, to compute the approximation ratio, one can perform a binary search over a cost relaxation parameter and, for each candidate value, test whether there exists a pair $T$ that is strictly improving for at least $\frac{2n}{k}$ agents. This procedure runs in polynomial time.

Regarding core testing, applying the same brute force protocol would require considering an exponential number of coalitions $T \subseteq C$. Instead, we test core stability via the following integer program, denoted CORETESTING.

$$\max \sum_{i \in N} x_i$$

$$\text{s.t.} \quad x_i \leq \sum_{T' \in \mathcal{P}_i(Y)} y_{T'} \qquad\qquad \forall i \in N$$

$$y_{T'} \leq y_s \qquad\qquad \forall T' \subseteq C, |T'| = 2, s \in T'$$

$$\sum_{i \in N} x_i \geq \frac{n}{k} \sum_{s \in C} y_s$$

$$x_i \in \{0,1\} \qquad\qquad \forall i \in N$$

$$y_s \in \{0,1\} \qquad\qquad \forall s \in C$$

$$y_{T'} \in \{0,1\} \qquad\qquad \forall T' \subseteq C, |T'| = 2$$

We then recall the Proposition C.3 by Bullinger et al. (2025) which shows how the integer program checks the core satisfaction for any given solution.

**Proposition F.2** ((Bullinger et al., 2025)). *A solution $Y$ is in the core if and only if its corresponding integer program (CORETESTING) has an optimal value of $0$.*

By Proposition F.2, we can again use binary search to identify the smallest cost relaxation parameter for which, under that parameter, the instance induced by $Y$ yields an integer program with optimal value $0$.

**Data and Sample.** We use trip records from the City of Helena Capital Transit Service dataset, which contains $10,282$ unique routes among citizens over $3,075$ distinct spatial points. Each record specifies a pick up point and a drop off point. To obtain the underlying travel costs, we combine these trip endpoints with OpenStreetMap road network data and use the open source Valhalla routing engine to compute shortest path travel costs. This allows us to derive both walking costs and shuttle bus transit costs between any pair of points in the spatial set.

Our experiments are designed to compare algorithmic performance across different stop numbers $k$ and cost scaling settings. For each parameter combination, we sample either $400$ agents or $40$ agents together with their associated routes, and take the union of their origin and destination points as the candidate facility location set. We repeat the sampling procedure $50$ times and present, for each algorithm, the average as well as the minimum and maximum approximation ratios across the sampled instances.

### F.2. JR Approximation Evaluation

We provide a more detailed comparison with respect to the following three algorithms, including the GC-TrSP, ECA, and $\frac{1}{2}$-Hybrid algorithm.

Our first observation is that all three algorithms achieve approximation ratios for JR that are close to $1$ across repeated random sampling. This suggests that, on average, each method returns (near) exact JR solutions under the tested regimes. Under lower transit cost scaling, ECA is consistently worse in expectation, although the performance gap remains modest.

In this regime, the $\frac{1}{2}$-Hybrid algorithm closely tracks GC-TrSP: both maintain averages very near 1 and exhibit only a mild upward trend as $k$ increases.

As the transit cost scaling increases, the behavior of ECA changes substantially. Its average approximation ratio decreases as $k$ grows, and its outcomes become more concentrated, indicating improved robustness with respect to JR approximation. In contrast, GC-TrSP and $\frac{1}{2}$-Hybrid display a gradual increase in their averages as $k$ increases. For sufficiently large $k$, the three methods largely converge to comparable approximation levels.

When the transit cost scaling reaches 10, that is, when walking or other alternative modes are viable for a substantial fraction of routes, ECA becomes the best performer on average and also exhibits the strongest robustness.

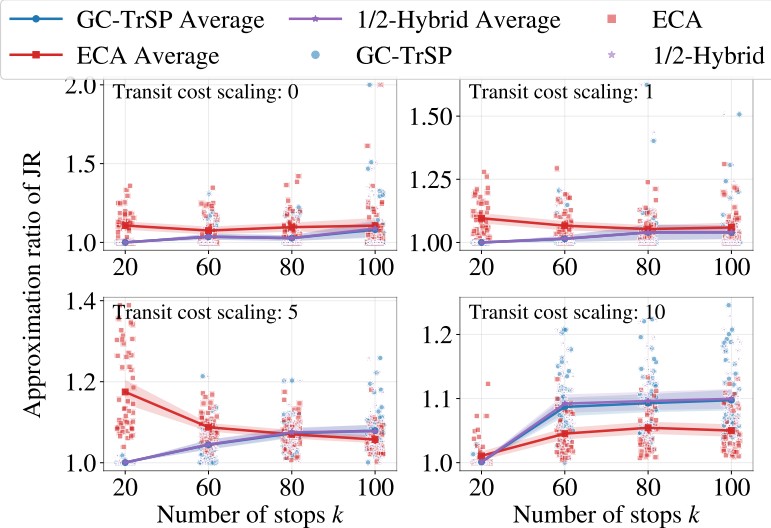

*Figure 9.* JR approximation evaluation for comparing GC-TrSP, ECA and $\frac{1}{2}$-Hybrid with 400 agents. Stop selection size ranges from 20 to 100, and the transit cost scale ranges from 0 to 10. Distribution of instance approximation ratios and the mean approximation ratio with 95% confidence intervals

### F.3. Core Approximation Evaluation

Figure 10 shows the empirical approximation ratio to the $(2, \beta)$-core achieved by GC-TrSP, ECA, and the $\frac{1}{2}$-Hybrid algorithm, as a function of the stop number $k$ and the cost relaxation parameter $\beta$ under different transit cost scaling regimes. Across all panels, the three methods behave almost identically for small $k$ (in particular $k = 2, 4$), with mean approximation ratios essentially equal to 1, indicating that the returned solutions typically lie in the exact core.

As $k$ increases, the methods begin to separate. ECA is consistently the most sensitive to larger $k$, exhibiting a clear upward drift in its average approximation ratio and substantially larger dispersion across instances, especially when transit costs are small. In contrast, GC-TrSP remains highly stable across all $k$, with averages staying very close to 1. The $\frac{1}{2}$-Hybrid algorithm interpolates between the two, tracking GC-TrSP closely while showing a modest increase for larger $k$, but with much less variance than ECA. While at scaling 10, all three algorithms are essentially indistinguishable, producing near exact core outcomes across all $k$. Notably, ECA admits an unbounded worst case core approximation ratio, yet in our experiments it remains well behaved, never exceeding an approximation ratio of 2.

## G. Transit Stop Placement on a Line

Although the algorithm by Bullinger et al. (2025) was originally designed for TrSP, it naturally extends to the clustering problem when the metric space is a line. Specifically, the algorithm processes the data sequentially from the leftmost point, selecting the nearest point on the right to form clusters of size $\lceil \frac{n}{k} \rceil$. We demonstrate that this algorithm satisfies PF when $\mathcal{C} = \mathcal{X}$.

**Proposition G.1.** *Algorithm 1 by Bullinger et al. (2025) satisfies PF in clustering on a line when $\mathcal{C} = \mathcal{X}$.*

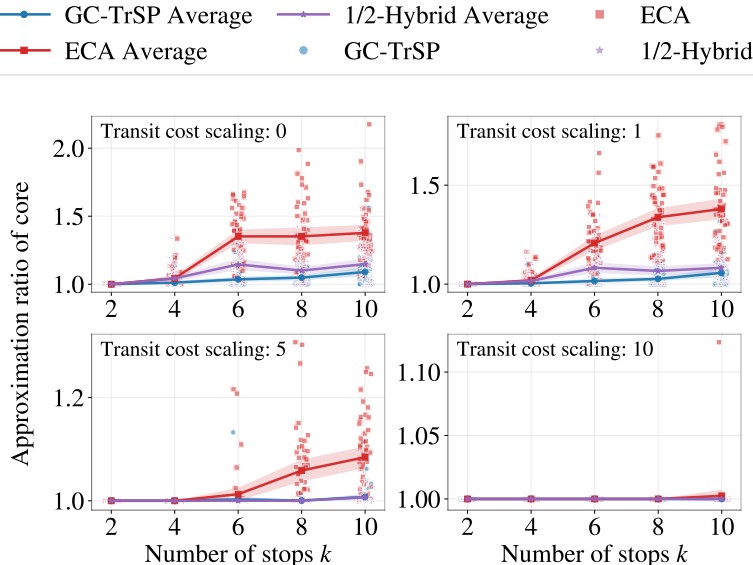

*Figure 10.* Core approximation evaluation for comparing GC-TrSP, ECA and $\frac{1}{2}$-Hybrid with 40 agents. Fixing the size relaxation parameter $\alpha$ as 2. Stop selection size ranges from 2 to 10, and the transit cost scale ranges from 0 to 10. Distribution of instance approximation ratios and the mean approximation ratio with 95% confidence intervals

*Proof.* Assume for contradiction that Algorithm 1 by Bullinger et al. (2025) does not satisfy PF, that is, for the solution $P$ by the algorithm, there exists a subset of agents $S \subseteq N$ of size $|S| \geq \lceil \frac{n}{k} \rceil$ and some center $y \in \mathcal{C}$ such that $d(i, y) < d(i, P)$ for every $i \in S$. Obviously, we have $y \notin P$. Now consider $y$ lies between any two centers $c$ and $c'$ in $P$. In this case, there will be at least $\lceil \frac{n}{k} \rceil$ data points lie strictly between $c$ and $c'$ as $d(i, y) < d(i, P)$ holds for every $i \in S$. However, according to the center selection by Algorithm 1 by Bullinger et al. (2025), there are at most $\lceil \frac{n}{k} \rceil - 1$ date points between $c$ and $c'$ (otherwise $c'$ cannot be selected as a center), which implies contradiction. For the case that $y$ lies on the left (right) side of the leftmost (rightmost) center in $P$, since there are at least $\lceil \frac{n}{k} \rceil$ agents on the left (right) side of leftmost (rightmost) center in $P$, a contradiction shows up as one more center should be selected by the design of the algorithm. □

Proposition G.1 establishes that Algorithm 1 by Bullinger et al. (2025) guarantees PF solutions for clustering on a line when the candidate set coincides with the set of agents, i.e., $\mathcal{C} = \mathcal{X}$. Combined with Theorem 3.1, Proposition G.1 immediately yields the upper bound of Theorem 4.4 of Bullinger et al. (2025), i.e., these statements show that Algorithm 1 by Bullinger et al. (2025) satisfies 2-core when $\mathcal{C} = \mathcal{X}$. However, when the candidate set is arbitrary (allowing agents to be distinct from candidates), the performance of their algorithm can deteriorate significantly. To illustrate this, we present the following example in Figure 11.

Consider an instance with $N = \{1, 2, 3, 4\}$, $k = 2$, and candidates $\{c_1, c_2, c_3, c_4\}$. All the data points and candidates are shown in Figure 11. According to the algorithm by Bullinger et al. (2025), candidates $c_2$ and $c_4$ will be selected as the

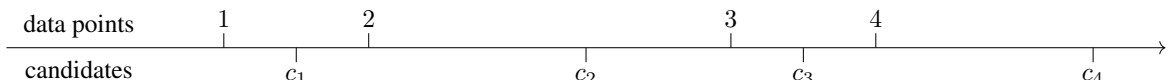

*Figure 11.* Example of Algorithm by Bullinger et al. (2025) fails PF

centers. However, agents $\{1, 2\}$ (resp. agents $\{3, 4\}$) form a deviation coalition that prefers center $c_1$ (resp. center $c_3$). The approximation ratio of PF can be arbtrarily poor as $d(1, c_1)$ and $d(2, c_1)$ can be arbtrarily small while $d(1, c_2)$ and $d(2, c_2)$ can be arbtrarily large.

As a fitting follow-up to this observation, we propose a novel algorithm named $\ell$-dictator partition algorithm, which ensures PF for general cases on a line.

**Theorem G.2.** *The $\ell$-dictator partition algorithm satisfies 1-PF in clustering on a line.*

---

**Algorithm 4** $\ell$-dictator partition algorithm

---

**Input:** TrSP instance $\mathcal{I} = \langle N, \mathcal{C}, k, \{\theta_i\}_{i \in N} \rangle$
**Output:** $P$.
1: Initialize $P \leftarrow \emptyset$, $\ell$ be a constant such that $\ell \leq \lfloor n/k \rfloor$.
2: **for** $i = 0, \ldots, k-1$ **do**
3:     Let $j$ be the $\ell$-th agent among $\{i\lceil n/k \rceil + 1, i\lceil n/k \rceil + 2, \ldots, (i+1)\lceil n/k \rceil\}$.
4:     $c_i \leftarrow \min_{c \in \mathcal{C} \setminus P} d(j, c)$.
5:     Update $P \leftarrow P \cup \{c_i\}$.
6: **end for**

---

*Proof.* Consider any arbitrary subset $S$, of agents with size at least $\lceil \frac{n}{k} \rceil$. For any subset of candidates $T$, we prove that there always exists an agent $i$ in $S$ such that $i$ prefers the clustering $P$ returned by the $\ell$-dictator partition algorithm to $T$, i.e., $\exists\, i \in S, d(i, P) < d(i, T)$. Without loss of generality, we denote the location of the leftmost and rightmost agents in $S$ by $\mathrm{lm}(S)$ and $\mathrm{rm}(S)$.

**Case 1:** There exists $c \in P$ such that $c \in (\mathrm{lm}(S), \mathrm{rm}(S))$.
Consider any unselected candidate $y \in \mathcal{C} \setminus P$. Assuming that $y$ is on the left (right) side of $c$, then there always exists at least one agent $i$ located on the right (left) side of $c$, meaning that $d(i, y) > d(i, c) \geq d(i, P)$.

**Case 2.** There is no candidate $c \in P$ satisfying $c \in (\mathrm{lm}(S), \mathrm{rm}(S))$.
First note that, due to the size of $S$, there must be some $\ell$-dictator agent in the interval $[\mathrm{lm}(S), \mathrm{rm}(S)]$. If $S$ contains some $\ell$-dictator agent, then $P$ contains this agent's closest center and we are done. We assume henceforth that there is instead some agent $q \in (\mathrm{lm}(S), \mathrm{rm}(S)) \setminus S$ who is an $\ell$-dictator.

The first sub-case is that all of the agents in $S$ are placed between two selected centers, denoted as $c$ and $c'$. It is apparent that $S$ would not deviate to a candidate $y$ outside of the interval between $c$ and $c'$, i.e. some candidate left of $c$ or right of $c'$. Hence, we focus on an arbitrary candidate $y$ in the interval between $c$ and $c'$. Notice that $y$ is not selected by agent $q$, implying that the distance from agent $q$ to $c$ ($c'$) is strictly smaller than that to $y$. Hence, there must exist at least one agent $i$ on the left (right) side of $q$ in $S$ who prefers $c$ ($c'$);

The second sub-case is that all of the agents in $S$ are placed on the left of the leftmost selected center or on the right of the rightmost selected center. Supposing all of the agents in $S$ are placed on the left of the leftmost selected center $c$, then $q$ always selects $c$. No agent in $S$ will deviate to any candidate center $y$ on the right side of $c$. Furthermore, there must not be a candidate center between the location of $q$ and $c$, since otherwise, $q$ would not select $c$. Lastly, note that the agent $\mathrm{rm}(S)$ would prefer $c$ to any candidate center $y$ left of $q$ (since $q$ also prefers $c$ to $y$). If the agents are on the right of the rightmost center, the statement holds by an analogous argument. This completes the proof. $\qquad\square$

