# OpenReview forum: "Fair Transit Stop Placement: A Clustering Perspective and Beyond"
_ICML.cc/2026/Conference — ICML 2026 regular_

### Official Review · Reviewer_czNm · 2026-03-09

**Soundness:** 3
**Presentation:** 3
**Significance:** 3
**Originality:** 3
**Overall Recommendation:** 5
**Confidence:** 3

**Summary:**

This paper studies a (let's call it) network design problem. There are agents in a metric space, each agent has a target and a destination and the goal is to build a fixed number of transportation hubs. The agents then use these hubs to go from their target to their destination. The goal is for this hub selection to be fair. Fairness in particular is interpreted as proportional fairness and core stability: no group of agents should be able to choose a subselection of transit stops proportional to their size, such that all the agents would be better off with this selection. Previous work has studied this model only in the one-dimensional setting. This paper extends this to general metric spaces. In particular, it is shown that if the transportation cost between hubs is 0, the setting is closely related to the proportional clustering setting, with proportionality in the clustering setting also implying proportionality in the bus stop design setting. If transportation costs are non-neglible, proportional outcomes can still be achieved by a "new" algorithm. Finally, both the clustering and the new algorithm are evaluated experimentally on a dataset constructed from real-world traffic data.

**Compliance With Llm Reviewing Policy:**

Affirmed.

**Final Justification:**

Solid paper, with interesting ideas. I particularly enjoyed the reduction to clustering. Will therefore (of course) stay as an accept.

**Key Questions For Authors:**

/

**Limitations:**

Yes

**Strengths And Weaknesses:**

I think I overall quite like this paper. While at heart seeming more like an OR than an ML paper, I still think it would be a good fit for ICML, in particular because of the connection to proportional clustering. Overall, the paper is quite well written (and for the most part easy to follow, even in the more technical proofs). These proofs at parts could be better presented, but overall, there is not too much to complain about. Overall, I think the paper is a nice contribution to a "cute" topic and improves upon previous work quite well (I have reviewed the Bullinger et al., paper for a previous venue before and the restriction to the line was something that needed to be improved upon). Thus, in summary, a nice and clean work, with a clear message and a good fit ICML in my eyes.

More detailed comments:

- "which we refer to as null transit times, captures scenarios where the cost of using the transit system is negligible compared to walking. Critically, our central algorithmic result will hold for arbitrary transit times." I think you could be (even) more transparent throughout the paper where you use the null transit assumption and where you dont. In particular, the statement of Theorem 3.1 should include the fact that it only works for null transfers.

- "Since solution Y" -> "Since the solution Y"

-  I dont really like the "three dot" part in the proof of Theorem 3.1, that is something which I would call an anti-pattern

- "implies that the solution Y satisfies ..." satisfies core sounds a bit wrong, should either be "is in the ()-core" or satisfies core-stability

- "core solutions for the TrSP problem can be computed using the clustering algorithm known as Greedy Capture "  should probably be highlighted that this is not polynomial-time (at least for Euclidean spaces)

- The statement of "Proposition 3.2." I think could be strengthened slightly. At present, it is still possible that for instance the (2,2)-core could be selected by GC-TrSP. So this is not exactly tight yet, as the second part could be improved.

- "Also, while Y could also contain" double also is a bit awkward

---

> ### Author Rebuttal · Authors · 2026-03-27
>
> Thank you for your encouraging review. We are glad that you found the problem we study interesting. We would like to highlight some of your comments in our rebuttal.
>
> ## Transparency about the transit cost assumption.
>
> Thank you for bringing this concern to our attention. We will make the null transit cost assumption more explicit in the next revision, in particular by specifying this assumption in every relevant statement. In Section 3, we will explicitly state the null transit cost assumption for the reductions between the TrSP problem and the clustering problem in both directions, as well as for the analysis of the Greedy Capture algorithm. In Section 4, we will clarify that all of our arguments and results hold under arbitrary transit cost assumptions.
>
> ## Statement regarding Proposition 3.2
>
> Thank you for raising this concern. We agree that this result is not exactly “tight”. We will soften the language in our treatment of this result.
>
> ## For other detailed minor comments.
>
> We are grateful for your detailed comments for our submission. We will make edits accordingly in the next revision.

---

> > ### Author Rebuttal · Reviewer_czNm · 2026-04-03
> >
> > Thanks!

---

### Official Review · Reviewer_j9yG · 2026-03-11

**Soundness:** 3
**Presentation:** 2
**Significance:** 3
**Originality:** 3
**Overall Recommendation:** 4
**Confidence:** 2

**Summary:**

This paper studies the transit stop placement (TrSP) problem in general metric spaces, where agents travel between source–destination pairs and can either walk directly or use a shuttle service via a selected set of transit stops. The paper investigates fairness in TrSP through the lenses of justified representation (JR) and the core, and reveals a structural connection to fair clustering. It shows that a constant-factor approximation to proportional fairness in clustering yields a constant-factor bi-parameter approximation to the core in TrSP. The authors also establish a lower bound of 1.366 on the approximability of JR and further show that no clustering-based algorithm can achieve a factor better than 3 for JR. To address this limitation, they propose the ECA algorithm, which obtains a tight 2.414-approximation for JR, and introduce a parameterized algorithm that interpolates between the two approaches, enabling a tunable trade-off between JR and the core. Finally, the paper complements its theoretical results with an experimental evaluation on small-market public carpooling data.

**Compliance With Llm Reviewing Policy:**

Affirmed.

**Final Justification:**

After reviewing the authors’ responses, I still maintain my 'weak accept' recommendation. However, the paper still needs improvements in presentation to fully meet the acceptance standard.

**Key Questions For Authors:**

1. Could the fairness framework proposed in this paper be extended to incorporate other notions of fairness, such as group fairness or individual fairness?
2. Could the authors provide a more detailed analysis of the time complexity of the algorithm?

**Limitations:**

The paper lacks the statement of impact, the authors should claim the impact of their results.

**Strengths And Weaknesses:**

Strengths:

1. The paper provides a strong theoretical results: approximation algorithms, impossibility results, existence lower bounds, and a trade-off between JR and core.
2. The ECA algorithm is novel and improves over all clustering-based approaches for JR, while also handling arbitrary transit cost functions.
3. The problem is well motivated and the fairness constraint is meaningful. Extending fair transit stop placement from the line to general metric spaces is a meaningful and nontrivial step.

Weakness:
1. The paper should be reorganized to improve clarity and readability. The interleaving of algorithm descriptions with lemmas and proofs makes it difficult to follow the main workflow and understand the core contributions.
2.The presentation is fairly dense, especially for readers who are not already familiar with proportional fairness and fair clustering.
3. The paper lacks a clear framework between their several theoretical results to make the readers understand.
4. The paper lacks the statement of impact, the authors should claim the impact of their results.
5. The paper lacks an analysis of the time complexity of its algorithms.

---

> ### Author Rebuttal · Authors · 2026-03-27
>
> Thank you for your insightful comments and suggestions for our submission.
>
> ## Q: Could the fairness framework proposed in this paper be extended to incorporate other notions of fairness, such as group fairness or individual fairness?
>
> Thank you for your question. The proportional fairness notions we considered in this paper, including JR and core, are specifically targeting group fairness. Critically, our fairness notions apply to arbitrary groups of agents, as opposed to a group fairness notion targeting pre-specified groups. Our algorithms implicitly find groups deserving representation using agents’ preferences. Algorithms targeting representation of pre-specified groups would want to additionally consider group membership in making stop selections. We are not certain what you mean by 'individual fairness’. If you mean individual fairness from the clustering literature [Jung et al. 2020], we agree it would be interesting to derive an analog of Theorem 3.1 for individual fairness, as well as other fairness notions from the clustering literature. We focused on proportional fairness because it has the most precedent in the literature and because it seems the most qualitatively similar to our notion of core.
>
> ## Q: Time complexity for the algorithms.
>
> Thank you for raising this question. We will include a more detailed analysis regarding the time complexity for the algorithms in our next revision. In particular, for the Greedy Capture (GC-TrSP) algorithm, the TrSP instance is essentially treated as a clustering instance, and its running time is $\tilde{O}(mn)$ time ([Chen et al. 2019]) where $m$ denotes the size of candidate transit stops and $n$ denotes the number of agents. For simplicity, we use $\tilde{O}(mn)$ to denote $O(mn)$ up to poly-logarithmic factors. Regarding our ECA algorithm, it first enumerates all pairs of candidate stops, which takes time $O(m^2)$. It then follows a procedure similar to that of Greedy Capture, for which we can again apply the discretization technique used in Greedy Capture, which gives us an overall running time of $\tilde{O}(m^2n)$.
>
> ## Missing Statement of Impact.
>
> Thank you for reminding us about the Statement of Impact section. We will include this in our next revision, highlighting the generality of our model and the potential impact of axioms and algorithms in both devising and evaluating provably fair transportation systems.
>
> 1. [Jung et al.,2020]. Jung, C., Kannan, S., & Lutz, N. (2020). Service in your neighborhood: Fairness in center location. Foundations of Responsible Computing (FORC), 5:1–5:15.
> 2. [Chen et al. 2019] Chen, X., Fain, B., Lyu, L. and Munagala, K. (2019) Proportionally fair clustering. International Conference on Machine Learning (ICML), 1032-1041.

---

> > ### Author Rebuttal · Reviewer_j9yG · 2026-04-05
> >
> > Thanks for the rebuttal. After reading the rebuttal and the other reviews, I will keep my score.

---

### Official Review · Reviewer_TMQc · 2026-03-12

**Soundness:** 3
**Presentation:** 3
**Significance:** 3
**Originality:** 3
**Overall Recommendation:** 5
**Confidence:** 3

**Summary:**

This paper studies the Transit Stop Placement (TrSP) problem under fairness constraints. The authors establish connections between fair clustering algorithms and the placement of transit stops in metric spaces. They propose several algorithms, including the Expanding Cost Algorithm (ECA) and a hybrid method that balances fairness guarantees related to justified representation and core stability.
Overall, this article discusses a central aspect of fairness-aware infrastructure design in transportation planning. Overall, the article examines algorithmic fairness in transit stop placement by drawing on theoretical connections to clustering algorithms.

**Compliance With Llm Reviewing Policy:**

Affirmed.

**Key Questions For Authors:**

1. How would the proposed algorithms scale to large real-world transit networks?
2. Can the authors relax the assumption of zero transit time between stops?
3. Are there datasets from urban transportation systems that could be used for evaluation?

**Limitations:**

Yes

**Strengths And Weaknesses:**

Strengths
● The problem is well-motivated and relevant to infrastructure planning.
● The paper provides strong theoretical contributions, including approximation guarantees and hardness results.
● The connection between fair clustering and transit stop placement is insightful.
● The proposed algorithms offer new theoretical perspectives on fairness.
Weaknesses
1. The empirical evaluation is limited and uses small datasets.
2. Several assumptions (e.g., null transit times) may reduce practical realism.
3. The fairness definitions may not fully capture real-world transportation equity concerns.
4. The scalability of the algorithms for large urban datasets is not analyzed.

---

> ### Author Rebuttal · Authors · 2026-03-27
>
> Thank you for your comprehensive and insightful comments and feedback.
>
> ## Q: How would the proposed algorithms scale to large real-world transit networks?
>
> As noted in our response to Reviewer j9yG, we would like to clarify that both the Greedy Capture algorithm and the ECA algorithm run in polynomial time. This means that they can be executed efficiently even on large real-world transit networks. For the evaluation, checking JR and computing the corresponding approximation ratio can also be done in polynomial time. In contrast, checking whether an outcome satisfies the core requires solving integer linear programs (ILPs), which is not a polynomial-time procedure. Therefore, verifying the core and computing the approximation ratio with respect to the core constitute the main implementation bottleneck for large real-world transit networks.
>
> ## Q: Can the authors relax the assumption of zero transit time between stops?
>
> We would like to emphasize that our main algorithmic contribution (Theorem 4.1) holds for arbitrary transit costs. The results in Section 3 assume null transit costs, and in several cases we provide examples showing that these results do not extend to the setting of arbitrary transit costs. We leave the problem of guaranteeing a constant-factor approximation to the core under general transit costs as an interesting open problem for future work.
>
> ## Q: Are there datasets from urban transportation systems that could be used for evaluation?
>
> Thank you for your question. In fact, any dataset containing per-user trip data (i.e., origin and destination points) together with a reasonable set of candidate stop locations could be used to evaluate the fairness of existing stop locations. It is worth noting that our method for checking core approximation requires solving many ILPs, which may become intractable for large-scale transportation systems. This contrasts with checking JR approximation factors, which is computationally efficient.

---

> > ### Author Rebuttal · Reviewer_TMQc · 2026-04-05
> >
> > Thanks for the answers. 1st question is not addressed adequately.

---

> > > ### Author Response · Authors · 2026-04-07
> > >
> > > We thank the reviewer for their comment. We apologize that, in our previous response, we interpreted the question on “scalability to large real-world transit networks” primarily in terms of computational tractability, which may not have fully captured the reviewer’s intended concern. If the reviewer is referring to a different notion of scalability, we would be very grateful for further clarification and would be happy to elaborate accordingly.

---

### Official Review · Reviewer_y1Mk · 2026-03-12

**Soundness:** 2
**Presentation:** 3
**Significance:** 3
**Originality:** 3
**Overall Recommendation:** 4
**Confidence:** 3

**Summary:**

The paper studies fair transit stop placement where riders have an origin-destination pair, and the algorithm designer selects $k$ transit stops. Rider's travel costs are then the minimum of walking directly between the two endpoints and taking the transit between two selected stops. The goal is to pick stops that is fair to all riders (subject to group level definitions of fairness). The paper proves approximate relationships between these fairness definitions, gives algorithms for the problem with constant approximation guarantees via mapping the problem onto fair clustering, and further provides lower bound constructions with various impossibility results.

**Compliance With Llm Reviewing Policy:**

Affirmed.

**Final Justification:**

I've increased my score since the my concerns with the proofs seem resolved. Keeping it as a weak accept because I did not check all appendix proofs and do worry that other minor errors exist and need resolved.

**Key Questions For Authors:**

Are my noted issues with the proofs valid and can they be easily rectified?

How is the JR approximation computed for the experiments? Did you search over all possible stop pairs and eligible groups?

**Limitations:**

The experiments are limited to one dataset and a very manufactured experimental setup to serve as a sanity check on the algorithms.

**Strengths And Weaknesses:**

### Strengths
The paper addresses a natural limitation of the prior work and makes interesting connections to the fair clustering problem. The prior work's focus on the line metric is very simplistic, and generating this set of results for a general metric is much needed to make these algorithms more useful in practice. The paper's presentation is largely clear and flows in an intuitive manner to help the reader grasp the technical novelty. All algorithmic results are presented with comparable lower bounds, though the question remains as to how tight each such result is.

### Weaknesses
The main weakness is potential bugs in the proofs, outlined below. I encourage the authors to respond to each and please let me know if I am mistaken!

**Theorem 3.1:** There appears to be an incorrect (or at least misspecified) application of the pigeonhole principle. $Q$ is defined to be a subset of $\Theta(S)$ and derive $|Q| > \frac{2n}{k} |T|$, and then there is an immediate application of the pigeonhole to conclude some agent in $S$ has both endpoints in $Q$. I do not think this following from the bound on $Q$ alone because $|S|$ can be much larger than $|T| \frac{2n}{k}$. The correct argument should require that $|Q| > |S|$.

**Lemma 3.4:** It is not clear how the defined value of $\epsilon$ gives us that $d(i, Y_b) \le \epsilon \cdot d(i,\tau)$. Should this bound not instead have $\frac{1}{\epsilon}$?

**Theorem 3.3:** The final max min taken to yield the $2 + \sqrt{5}$ seems to have an inconsistency. Its not immediately clear that you can introduce the variable $q$ as a stand-in for the dimensionless $r_T$ (which is already its own maximizer) and distance term $x$. I believe this proof result should still hold but the argument needs to introduce $q$ as a dimensionless variable that is independent of the scaling of $x$ (and equivalently $y$. From there you should be able to do $\max_{q > 0} \min$ ...

**Metric Issue (broadly):** several of the impossibility results require infinite distance between points which seems to violate the metric model we are working in. I think just replacing the infinity by some constant that is sufficiently large will make the arguments correct, but in the current form I do not think these results are technically correct.

---

> ### Author Rebuttal · Authors · 2026-03-27
>
> Thank you for your comprehensive and insightful comments on our paper. We would like to address and clarify the questions and concerns you raised.
>
> ## Q: An incorrect (or at least misspecified) application of the pigeonhole principle in Theorem 3.1.
>
> Thank you for bringing this to our attention. Indeed, there is a minor bug in the proof and it can be rectified by instead bounding $|Q| > |\Theta(S)| - |T| 2n/k \geq |\Theta(S)| -  ½|\Theta(S)| = ½|\Theta(S)|$, which follows from $|T| 2n/k \leq ½|\Theta(S)|$, a fact noted earlier in the proof. Then, the application of the pigeonhole principle is sufficient to show that some agent in $S$ has both endpoints in $\Theta(S)$.
>
> ## Q: It is not clear how the defined value of $\varepsilon$ gives us that $d(i, Y_b) \leq \varepsilon\cdot d(i,\tau)$.
>
> Thank you for your question. We would like to clarify that there is no bug in this proof. Please note that we actually want to show the opposite inequality in our proof, i.e., $d(i,Y_b) \geq \epsilon\cdot d(i,\tau)$. This then implies that $(2\beta - \epsilon) d(i,\tau) \geq 2\beta d(i,\tau) - d(i,Y_b)$. To see why $d(i,Y_b) \geq \epsilon\cdot d(i,\tau)$, note that $\epsilon = \frac{\min_{i’\in N’, \tau’\in C’} d(i’,\tau’)} {\max_{i'\in N’, \tau'\in C’} d(i',\tau') }\leq \frac{d(i,Y_b)} {d(i,\tau)}.$
>
> ## Q: Inconsistency issue in the proof of Theorem 3.3.
>
> Thank you for bringing this to our attention. We were unnecessarily optimizing over $r_T$, $x$, and $y$, generating inconsistencies. In fact, we can significantly simplify the sequence of inequalities as follows:
>
> $$
> \min\left( \frac{2\cdot r_T  + y}{x}, \frac{2\cdot r_T + x}{y}, \frac{3\cdot r_T + x + y}{r_T} \right) \leq
> \min\left( \frac{2r_T+x}{x}, 3+\frac{2x}{r_T} \right) \leq
> \max_{q \geq 1} \left(\min\left( 2q+1, 3 + 2/q \right)\right).
> $$
>
> All that remains is to justify why we can assume $x=y$, yielding the first inequality above. First note that if $x=0$ or $y=0$, then our inequality gives a bound even better than $2+\sqrt{5}$ so we can restrict to $x,y>0$. We now observe that $\frac{2z+x}{x}\geq \min(\frac{2z+y}{x}, \frac{2z+x}{y})$ for any $x,y>0$, which we will now prove rigorously. We divide into two cases. (1). When $y\leq x$, since $x$ and $y$ are positive, we have that $\frac{2z+x}{x} - \frac{2z+y}{x}=\frac{x-y}{x}\geq 0$, which implies $\frac{2z+x}{x} \geq \min(\frac{2z+y}{x}, \frac{2z+x}{y})$. (2). When $y \geq x$, we immediately have $\frac{2z+x}{x} \geq \frac{2z+x}{y}$ as $y\geq x$, which further implies $\frac{2z+x}{x} \geq \min(\frac{2z+y}{x}, \frac{2z+x}{y})$. This explains the first inequality. The second inequality follows by setting $q=r_T/x$ (as in previous proofs) and we must constrain $q\geq 1$ since $r_T\geq x$ by definition.
>
> We will update our proof to reflect the argument above. If this does not fully resolve your concerns about the argument, please let us know and we are happy to further address them.
>
> ## Q: Metric issues regarding the use of infinite distances.
>
> Thank you for raising this issue. We agree that a distance of infinity does not fit within our metric model. Actually, this follows the precedent in the literature to use infinity to denote a sufficiently large constant (see, e.g., [Chen et al. 2019]). To be more rigorous, we will explicitly clarify this in the paper and rephrase any uses of “infinite” in plain language.
>
> ## Q: How is the JR approximation computed for the experiments?
>
> Thank you for your question. The details regarding how to compute the approximation ratio of JR are provided in the Appendix F.1 (line 1481-1491). For the sake of completeness, we summarize here, noting that our procedure follows the idea of [Bullinger et al. 2025]. We do enumerate all possible pairs of stops, however, it is unnecessary to enumerate all possible groups of agents. The reason is that given any pair of stops, we can identify the number of agents who have incentive to deviate to the pair instead of the solution. A solution fails JR if and only if there exists some pair which more than $\frac{2n}{k}$ agents prefer. To further compute the minimum approximation ratio, we employ a simple binary search algorithm that searches over the relaxation parameter and runs the aforementioned algorithm for checking JR. (If you are interested in the implementation code, we refer you to the function `compute_JR_approximation` in our source code file `func.py` in the supplementary material.)
>
> 1. [Chen et al. 2019] Chen, X., Fain, B., Lyu, L. and Munagala, K. (2019) Proportionally fair clustering. International Conference on Machine Learning (ICML), 1032-1041.
> 2. [Bullinger et al. 2025] Bullinger, M., Elkind, E., & Latifian, M. (2025). Towards Fair and Efficient Public Transportation: A Bus Stop Model. International Conference on Autonomous Agents and Multiagent Systems (AAMAS), 427-435.

---

> > ### Author Rebuttal · Reviewer_y1Mk · 2026-04-03
> >
> > Great, thanks! Happy to increase my score.

---

### Decision · Program_Chairs · 2026-04-30

**Decision:**

Accept (regular)

**Comment:**

The paper studies the Transit Stop Placement problem under fairness constraints. This is a network design problem where agents in a metric space have a target and a destination, and the goal is to open transportation hubs satisfying proportional fairness and core stability. The authors propose an algorithm for this problem in general metric spaces, extending previous results that concerned only the line metric. The authors experimentally evaluate their algorithms using real traffic data.
All reviewers agreed (including myself) that this is a well-motivated and solid paper that makes progress on an important problem. During the discussion, some concerns were raised regarding the correctness of certain proofs; however, all of them were resolved during the author response phase.